# On Linear Stability of SGD and Input-Smoothness of Neural Networks

**Chao Ma**
Department of Mathematics
Stanford University
Stanford, CA 94305
chaoma@stanford.edu

**Lexing Ying**
Department of Mathematics
Stanford University
Stanford, CA 94305
lexing@stanford.edu

## Abstract

The multiplicative structure of parameters and input data in the first layer of neural networks is explored to build connection between the landscape of the loss function with respect to parameters and the landscape of the model function with respect to input data. By this connection, it is shown that flat minima regularize the gradient of the model function, which explains the good generalization performance of flat minima. Then, we go beyond the flatness and consider high-order moments of the gradient noise, and show that Stochastic Gradient Descent (SGD) tends to impose constraints on these moments by a linear stability analysis of SGD around global minima. Together with the multiplicative structure, we identify the Sobolev regularization effect of SGD, i.e. SGD regularizes the Sobolev seminorms of the model function with respect to the input data. Finally, bounds for generalization error and adversarial robustness are provided for solutions found by SGD under assumptions of the data distribution.

## 1 Introduction

Stochastic gradient descent (SGD) is the most widely used optimization algorithm to train neural networks [25, 4]. By taking mini-batches of training data instead of all the data in each iteration, it was firstly designed as a substitute of the gradient descent (GD) algorithm to reduce its computational cost. Extensive researches are conducted on the convergence of SGD, both on convex and non-convex objective functions [21, 22, 5, 3]. In these studies, convergence is usually proven in the cases where the learning rate is sufficiently small, hence the gradient noise is small. In practice, however, SGD is preferred over GD not only for the low computational cost, but also for the implicit regularization effect that produces solutions with good generalization performance [7, 16]. Since a trajectory of SGD tends to that of GD when the learning rate goes to $0$, this implicit regularization effect must come from the gradient noise induced by mini-batch and a moderate learning rate.

When studying the gradient noise of SGD, a majority of work treat SGD as an SDE, and studies the noise of the SDE [15, 20, 17, 18, 32]. However, SGD is close to SDE only when the learning rate is small, and it is unclear that in practical setting whether the Gaussian noise of SDE can fully characterize the gradient noise of SGD. Some work resort to heavy-tailed noise like the Levy process [27, 37]. Another perspective to study the behavior of SGD is by its linear stability [30, 12]. This is relevant when the learning rate is not very small. The linear stability theory can explain the fast escape of SGD from sharp minima. The escape time derived from this theory depends on the logarithm of the barrier height, while the escape time derived from the diffusion theory based on SDE depends exponentially on the barrier height [32]. The former is more consistent with empirical observations [30].

35th Conference on Neural Information Processing Systems (NeurIPS 2021).

An important observation that connects the generalization performance of the solution with the landscape of the loss function is that flat minima tend to generalize better [13, 31]. SGD is shown to pick flat minima, especially when the learning rate is big and the batch size is small [16, 15, 30]. Algorithms that prefer flat minima are designed to improve generalization [6, 14, 19, 35]. On the other hand, though, the reason why flat minima generalize better is still unclear. Intuitive explanations from description length or Bayesian perspective are provided in [13] and [23]. In [10], the authors show sharp minimum can also generalize by rescaling the parameters at a flat minimum. Hence, flatness is not a necessary condition of good generalization performance, but it is still possible to be a sufficient condition. In the study of linear stability in [30], besides the sharpness (a quantity inversely proportional to flatness), another quantity named non-uniformity is proposed which roughly characterizes the second order moment of the gradient noise. It is shown that SGD selects solutions with both low sharpness and low non-uniformity.

In this paper, we build a complete theoretical pipeline to analyze the implicit regularization effect and generalization performance of the solution found by SGD. Our starting points are the following two questions: (1) *Why SGD finds flat minima?* (2) *Why flat minima generalize better?* Our answers to these two questions go beyond the flatness and cover the non-uniformity and higher-order moments of the gradient noise. This distinguishes SGD from GD, and is out of the scope that can be explained by SDE. For the first question, we extend the linear stability theory of SGD from the second-order moments of the iterator of the linearized dynamics to the high-order moments. At the interpolation solutions found by SGD, by the linear stability theory, we derive a set of accurate upper bounds of the gradients' moment. For the second question, using the multiplicative structure of the input layer of neural networks, we show that the upper bounds obtained in the first step regularize the Sobolev seminorms of the model function. Finally, bridging the two components, our main result is a bound of generalization error under some assumptions of the data distribution. The bound works well when the distribution is supported on a low-dimensional manifold (or a union of low-dimensional manifolds). An informal statement of our main result is

(**Main result**) *Around an interpolation solution of the neural network model, assume that (1)the $k$-th order moment of SGD's iterator of the linearized dynamics is stable, (2)with probability at least $1 - \varepsilon$ the testing data is close to a training data with distance smaller than $\delta$, and (3) both the model function and the target function are upper bounded by a constant $M$. Then, at this interpolation solution we have*

$$generalization\ error \lesssim n^{\frac{1}{k}}\delta^2 + \varepsilon M^2,$$

*where $n$ is the number of data. The bound depends on the learning rate and the batch size of SGD as constant factors.*

The formal description is stated in Theorem 6 of Section 5. Our analysis also provide bounds for adversarial robustness. As a byproduct, we theoretically show that flatness of the minimum controls the gradient norm of the model function at the training data. Therefore, searching for flat minima has the effect of Lipschitz regularization, which is shown to be able to improve generalization [24].

Lying at the center of our analysis is the multiplicative structure of neural networks, i.e. in each layer the output features from the previous layer is multiplied with a parameter matrix. This structure is a rich source of implicit regularization. In this paper, we focus on the input layer, and build connection between the model function's derivative with respect to the parameters and its derivatives with respect to the data. Concretely, let $f(\mathbf{x}, W)$ be a neural network model, with $\mathbf{x}$ being the input data and $W$ being the parameters. We split the parameters by $W = (W_1, W_2)$, where $W_1$ is the parameter matrix of the first layer, and $W_2$ denotes all other parameters. Then, the neural network can be represented by the form $f(\mathbf{x}, W) = \tilde{f}(W_1\mathbf{x}, W_2)$ due to the multiplicative structure of $W_1$ and $\mathbf{x}$. Accordingly $\nabla_{W_1}\tilde{f}(W_1\mathbf{x}, W_2)$ and $\nabla_{\mathbf{x}}\tilde{f}(W_1\mathbf{x}, W_2)$ take the form

$$\nabla_{W_1}\tilde{f}(W_1\mathbf{x}, W_2) = \frac{\partial f(W_1\mathbf{x}, W_2)}{\partial(W_1\mathbf{x})}\mathbf{x}^T, \qquad \nabla_{\mathbf{x}}\tilde{f}(W_1\mathbf{x}, W_2) = W_1^T\frac{\partial f(W_1\mathbf{x}, W_2)}{\partial(W_1\mathbf{x})}, \qquad (1)$$

where $\frac{\partial f(W_1\mathbf{x}, W_2)}{\partial W_1\mathbf{x}}$ is usually a long expression produced by back propagation from the output to the first layer. By (1) we have

$$\|\nabla_{\mathbf{x}}\tilde{f}(W_1\mathbf{x}, W_2)\|_2 \leq \frac{\|W_1\|_2}{\|\mathbf{x}\|_2}\|\nabla_{W_1}\tilde{f}(W_1\mathbf{x}, W_2)\|_2. \qquad (2)$$

Hence, $\nabla_{\mathbf{x}}\tilde{f}(W_1\mathbf{x}, W_2)$ is upper bounded by $\nabla_{W_1}\tilde{f}(W_1\mathbf{x}, W_2)$, given that $W_1$ is not too big and $\mathbf{x}$ is not too small. By the bound (2) we can derive the regularization effect of flatness at interpolation

solutions. To see this, let $\{(\mathbf{x}_i, y_i)\}_{i=1}^{n}$ be the training data, and $\hat{L}(W) := \frac{1}{2n} \sum_{i=1}^{n} (f(\mathbf{x}_i, W) - y_i)^2$ be the empirical risk given by the square loss. Let $W^* = (W_1^*, W_2^*)$ be an interpolation solution, i.e. $f(\mathbf{x}_i, W^*) = y_i$ for any $i = 1, 2, ..., n$, and $\hat{L}(W^*) = 0$. Define the flatness of the minimum as the sum of the eigenvalues of $\nabla^2 \hat{L}(W^*)$, i.e. flatness$(W^*) = \text{Tr}(\nabla^2 \hat{L}(W^*))$. $W^*$ being the interpolation solution implies

$$\text{flatness}(W^*) = \frac{1}{n} \sum_{i=1}^{n} \|\nabla_W f(\mathbf{x}_i, W^*)\|^2. \tag{3}$$

Hence, from (2) we can obtain the following equation that gives bounds for the gradients of the model function with respect to the input data at the training data,

$$\frac{1}{n} \sum_{i=1}^{n} \|\nabla_{\mathbf{x}} f(\mathbf{x}_i, W^*)\|^2 \leq \frac{\|W_1^*\|_2^2}{\min_i \|\mathbf{x}_i\|_2^2} \frac{1}{n} \sum_{i=1}^{n} \|\nabla_{W_1} f(\mathbf{x}_i, W^*)\|_2^2 \leq \frac{\|W_1^*\|_2^2}{\min_i \|\mathbf{x}_i\|^2} \text{flatness}(W^*). \tag{4}$$

The left hand side of (4) is usually used to regularize the Lipschitz constant of the model function, and such regularization can improve the generalization performance and adversarial robustness of the model. Hence, (4) reveals the regularization effect of flat minima. Later in the paper, we extend the analysis to higher-order moments of the gradient and combine the results with the linear stability theory of SGD to explain the implicit regularization effect of SGD. We also extend the bound on $\nabla_{\mathbf{x}} f(\mathbf{x}, W^*)$ from training data to all $\mathbf{x}$ in a neighborhood of the training data, which implies the regularization of flatness is actually stronger than the left-hand-side of (4).

The strong correlation between $\nabla_W f(\mathbf{x}, W)$ and $\nabla_{\mathbf{x}} f(\mathbf{x}, W)$ is justified by numerical experiments in practical settings. Specifically, in experiments we compare the following two quantities:

$$g_W := \left( \frac{1}{n} \sum_{i=1}^{n} \|\nabla_W f(\mathbf{x}_i, W)\|_2^2 \right)^{\frac{1}{2}}, \quad g_{\mathbf{x}} := \left( \frac{1}{n} \sum_{i=1}^{n} \|\nabla_{\mathbf{x}} f(\mathbf{x}_i, W)\|_2^2 \right)^{\frac{1}{2}}. \tag{5}$$

Figure 1 shows the results for a fully-connected network trained on FashionMNIST dataset and a VGG-11 network trained on CIFAR10 dataset. In each plot, $g_W$ and $g_{\mathbf{x}}$ of different solutions found by SGD are shown by scatter plots. The plots show strong correlations between the two quantities. The colors of the points show that SGD with big learning rate and small batch size tends to converge to solutions with small $g_W$ and $g_{\mathbf{x}}$, which is consistent with our theoretical results on the Sobolev regularization effect of SGD. To summarize, the main contributions of this paper are:

1. We extend the linear stability analysis of SGD to high-order moments of the iterators. At the solutions selected by SGD, we find a class of conditions satisfied by the gradients of different training data. These conditions cover the flatness and non-uniformity, and also include higher order moments. They characterize the regularization effect of SGD beyond GD and SDE.

2. By exploring the multiplicative structure of the neural networks' input layer, we build relations between the model function's derivatives with respect to the parameters and with respect to the inputs. By these relations we turn the conditions obtained for SGD into bounds of different Sobolev (semi)norms of the model function. In particular, we show that flatness of the minimum regularizes the $L^2$ norm of the gradient of the model function. This explains how the flatness (as well as other stability conditions for SGD) benefits generalization and adversarial robustness.

3. Still using the multiplicative structure, the bounds for Sobolev seminorms can be extended from the training data to a neighborhood around the training data, based on certain smoothness assumption of the model function (with respect to parameters). Then, bounds for generalization error and adversarial robustness are provided under reasonable assumptions of the data distribution. The bounds work well when the data are distributed effectively in a set that consists of low dimensional manifolds.

## 2 Preliminaries

**Basic notations** For any vector $\mathbf{x} = (x_1, ..., x_n)^T \in \mathbb{R}^n$ and any $p \geq 1$, $\|\mathbf{x}\|_p$ is the conventional $p$-norm of $\mathbf{x}$. If $g(\mathbf{x}) : \mathbb{R}^d \to \mathbb{R}^n$ is a function with vector output, $\|g(\mathbf{x})\|_p$ means the $p$-norm of the vector $g(\mathbf{x})$, instead of a function norm. For any matrix $A \in \mathbb{R}^{m \times n}$, $\|A\|_p$ is the operator norm

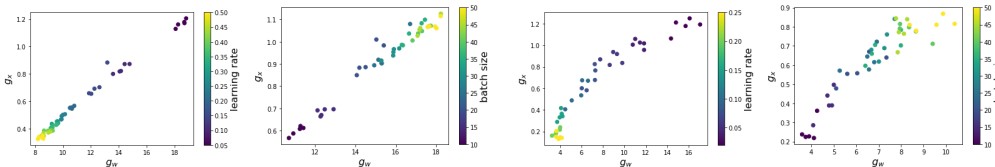

(a)                                                    (b)

Figure 1: **(a)** A fully-connected neural network trained on FashionMNIST. **(b)** A VGG-11 network trained on CIFAR10. For both (a) and (b), the left panel shows $g_W$ and $g_{\mathbf{x}}$ of the solutions found by SGD with different learning rate, while batch size fixed at 20. The right panel shows solutions found by SGD with different batch size, with learning rate fixed at 0.1.

induced by the $p$-norm on $\mathbb{R}^n$ and $\mathbb{R}^m$. $\|A\|_F$ denotes the Frobenius norm. Let $g : \mathbb{R}^n \to \mathbb{R}$ be a function, $\Omega \subset \mathbb{R}^n$ be a set in which $g$ is defined, $q \in \mathbb{N}^*$, and $p \geq 1$. The Sobolev seminorm $|g|_{q,p,\Omega}$ is defined as $|g|_{q,p,\Omega} = (\sum_{|\alpha|=q} \int_\Omega |D^\alpha g(\mathbf{x})|^p d\mathbf{x})^{\frac{1}{p}}$, where $\alpha = (\alpha_1, ..., \alpha_n)$ is a multi-index with $n$ positive integers, $|\alpha| = \sum_{i=1}^n \alpha_i$, and $D^\alpha g$ is defined as $D^\alpha g = \frac{\partial^{|\alpha|} g}{\partial x_1^{\alpha_1} \cdots \partial x_n^{\alpha_n}}$. The index $q$ can be extended to fractions, but in this paper we always consider integers. When $\Omega = \{\mathbf{x}_1, ..., \mathbf{x}_n\}$ is a finite set, we define the Sobolev seminorm as $|g|_{q,p,\Omega} = (\sum_{|\alpha|=q} \frac{1}{n} \sum_{i=1}^n |D^\alpha g(\mathbf{x}_i)|^p)^{\frac{1}{p}}$.

For a symmetric matrix $A \in \mathbb{R}^{n \times n}$, we let $\Lambda(A)$ be the set of eigenvalues of $A$ and $\lambda_{\max}(A)$ be the maximum eigenvalue of $A$. For two matrices $A$ and $B$, $A \otimes B$ denotes the Kronecker product of $A$ and $B$. For $k \in \mathbb{N}^*$, let $A^{\otimes k} = A \otimes A \otimes \cdots \otimes A$ where $A$ is multiplied for $k$ times. Therefore, if $A \in \mathbb{R}^{m \times n}$, then $A^{\otimes k} \in \mathbb{R}^{m^k \times n^k}$. However, for a vector $\mathbf{x} \in \mathbb{R}^n$, with an abuse of notation sometimes we also use $\mathbf{x}^{\otimes k}$ to denote the rank-one tensor product. In this case, $\mathbf{x}^{\otimes k} \in \mathbb{R}^{n \times n \times \cdots \times n}$.

**Problem settings**  In this paper, we consider the learning of the parameterized model $f(\mathbf{x}, W)$, with $\mathbf{x}$ being the input data and $W$ being the parameters. Let $d$ be the dimension of the input data and $w$ be the number of parameters in $W$, i.e. $\mathbf{x} \in \mathbb{R}^d$ and $W \in \mathbb{R}^w$. We assume that the model has scalar output. We consider the models with multiplicative structure on input data and part of the parameters. With an abuse of notation let $W = (W_1, W_2)$, where $W_1 \in \mathbb{R}^{m \times d}$ is the part of the parameters multiplied with $\mathbf{x}$ and $W_2$ denotes other parameters. Then, the model can be written as

$$f(\mathbf{x}, W) = \tilde{f}(W_1 \mathbf{x}, W_2). \tag{6}$$

We remark that most neural networks have form (6), including convolutional networks, residual networks, recurrent networks, and even transformers. Note that since the $\mathbf{x}$ in (6) can also be understood as fixed features calculated using input data, (6) also includes random feature models.

We consider a supervised learning setting. Let $\mu$ be a data distribution supported within $\mathbb{R}^d$, and $f^* : \mathbb{R}^d \to \mathbb{R}$ be a target function. A set of $n$ training data $\{(\mathbf{x}_i, y_i)\}_{i=1}^n$ is obtained by sampling $\mathbf{x}_1, ..., \mathbf{x}_n$ i.i.d. from $\mu$, and letting $y_i = f^*(\mathbf{x}_i)$. As mentioned in the introduction, the model is learned by minimizing the empirical loss function $\hat{L}(W) = \frac{1}{2n} \sum_{i=1}^n (f(\mathbf{x}_i, W) - y_i)^2$ in the space of parameters. The population loss, or the generalization error, is defined as $L(W) = \int (f(\mathbf{x}, W) - f^*(\mathbf{x}))^2 d\mu(\mathbf{x})$.

For any $i = 1, 2, ..., n$, let $L_i(W) = \frac{1}{2}(f(\mathbf{x}_i, W) - y_i)^2$ be the loss at $\mathbf{x}_i$. Then, the iteration scheme of the SGD with learning rate $\eta$ and batch size $B$ is

$$W_{t+1} = W_t - \frac{\eta}{B} \sum_{i=1}^B \nabla L_{\xi_i^t}(W_t), \tag{7}$$

where $\xi^t = (\xi_1^t, \xi_2^t, ..., \xi_B^t)$ is a $B$-dimensional random variable uniformly distributed on the $B$-tuples in $\{1, 2, ..., n\}$ and independent with $W_t$. In the paper, we study interpolation solutions $W^*$ found by SGD, which satisfies $f(\mathbf{x}_i, W^*) = y_i$ for any $i = 1, 2, ..., n$. Obtaining interpolation solutions is possible in the over-parameterized setting [34], and is widely studied in existing work [36, 33].

Some assumptions will be made when deriving the generalization error bounds. Firstly, we assume the model function with respect to the parameter $W$ to be smooth around $W^*$.

**Definition 1.** Let $\delta, C$ be positive numbers, and $k$ be a positive integer. We say the model $f(\mathbf{x}, W)$ satisfies the $(C, \delta, k)$-**local smoothness condition** at data $\mathbf{x}$ and parameter $W$, if for any $W'$ such

that $\|W' - W\|_2 \leq \delta$, there is

$$\|\nabla_W f(\mathbf{x}, W')\|_{2k} \leq C(\|\nabla_W f(\mathbf{x}, W)\|_{2k} + 1). \tag{8}$$

**Remark 1.** *In the definition above, we consider the $2k$-norm of the gradients for the convenience of later analysis. When $k = 1$, the condition (8) becomes $\|\nabla_W f(\mathbf{x}, W')\|_2 \leq C(\|\nabla_W f(\mathbf{x}, W)\|_2 + 1)$. This is weaker than local approximation by Taylor expansion, which usually yields results like $\|\nabla_W f(\mathbf{x}_i, W) - \nabla_W f(\mathbf{x}_i, W^*)\|_2 \leq C\delta$. Here, we only require that the gradient with respect to the parameters does not get exceedingly large when $W$ is close to $W^*$.*

Next, we need the data distribution $\mu$ to support roughly on a low dimensional manifold (or a union of low dimensional manifolds), thus the neighborhoods of training data can well cover all test data.

**Definition 2.** Let $\mu$ be a probability distribution supported in $\mathbb{R}^d$, and $\{\mathbf{x}_i\} = \{\mathbf{x}_1, ..., \mathbf{x}_n\}$ be a set of points in $\mathbb{R}^d$. For positive constants $\delta$ and $\varepsilon$, we say $\{\mathbf{x}_i\}$ $(\delta, \varepsilon)$-**covers** $\mu$ if

$$\mathbb{P}_{\mathbf{x} \sim \mu} \left( \min_{1 \leq i \leq n} \|\mathbf{x} - \mathbf{x}_i\|_2 > \delta \right) < \varepsilon, \tag{9}$$

i.e. with high probability a point sampled from $\mu$ lies close to a point in $\{\mathbf{x}_i\}$.

**Definition 3.** Let $\{\mathbf{x}_i\} = \{\mathbf{x}_1, ..., \mathbf{x}_n\}$ be a set of $n$ points in $\mathbb{R}^d$. For positive constants $\delta$ and $K$, we say $\{\mathbf{x}_i\}$ is $(\delta, K)$-**scattered** if the function $\kappa(\mathbf{x}) := \sum_{i=1}^{n} \mathbf{1}_{B(\mathbf{x}_i, \delta)}(\mathbf{x})$ satisfies $\kappa(\mathbf{x}) \leq K$. Here, $B(\mathbf{x}_i, \delta)$ is the ball in $\mathbb{R}^d$ centered at $\mathbf{x}_i$ with radius $\delta$, and $\mathbf{1}_A(\mathbf{x})$ is the indicator function of set $A$.

**Remark 2.** *The uniform upper bound $\kappa(\mathbf{x}) \leq K$ in the above definition can be weakened as an integral upper bound $\frac{1}{V_{\mathcal{X}}} \int_{\mathcal{X}} \kappa(\mathbf{x})d\mathbf{x} \leq K$, where $\mathcal{X} = \bigcup_{i=1}^{n} B(\mathbf{x}_i, \delta)$ and $V_{\mathcal{X}}$ is the Lebesgue volume of $\mathcal{X}$.*

**Definition 4.** Let $\mu$ be a probability distribution supported in $\mathbb{R}^d$. For positive integer $n$ and positive constants $\delta, \varepsilon_1, \varepsilon_2$, we say $\mu$ satisfies the $(n, \delta, \varepsilon_1, \varepsilon_2)$-**covered condition**, if with probability at least $1 - \varepsilon_1$ over the choice of $n$ i.i.d. sampled data from $\mu$, $\{\mathbf{x}_1, ..., \mathbf{x}_n\}$, we have

$$\mathbb{P}_{\mathbf{x} \sim \mu} \left( \min_{1 \leq i \leq n} \|\mathbf{x} - \mathbf{x}_i\|_2 > \delta \right) < \varepsilon_2. \tag{10}$$

On the other hand, for positive integer $n$ and positive constants $\delta$, $\varepsilon_1$ and $K$, we say $\mu$ satisfies the $(n, \delta, \varepsilon_1, K)$-**scattered condition**, if with probability at least $1 - \varepsilon_1$ over the choice of $n$ i.i.d. sampled data $\{\mathbf{x}_i\}$ from $\mu$, $\{\mathbf{x}_i\}$ is $(\delta, K)$-scattered.

Later in the analysis of the generalization error (Section 5), we will assume the model and the data distribution satisfy the conditions in Definition 1 and 4 with appropriately chosen constants.

## 3 Linear stability theory of SGD

Compared with full-batch GD, SGD adds in each iteration a random "noise" to the gradient of the loss function. Hence, it is harder for SGD to be stable around minima than GD. This is shown by the linear stability theory of SGD.

Recall that the iteration scheme of SGD with learning rate $\eta$ and batch size $B$ is given by (7). Let $W^*$ be an interpolation solution for the learning problem. When $\{W_t\}$ is close to $W^*$, the behavior of (7), including its stability around the minimum, can be characterized by the linearized dynamics at $W^*$:

$$\tilde{W}_{t+1} = \tilde{W}_t - \frac{\eta}{B} \sum_{i=1}^{B} \nabla_W f(\mathbf{x}_{\xi_i^t}, W^*) \nabla_W f(\mathbf{x}_{\xi_i^t}, W^*)^T (\tilde{W}_t - W^*). \tag{11}$$

The linearization is made by considering the quadratic approximation of $L_i$ near $W^*$. For ease of notation, let $\boldsymbol{a}_i = \nabla_W f(\mathbf{x}_i, W^*)$, $H_i = \boldsymbol{a}_i \boldsymbol{a}_i^T$. In the linearized dynamics (11), without loss of generality we can set $W^* = 0$ by replacing $\tilde{W}_t$ by $\tilde{W}_t - W^*$. Then (11) becomes

$$\tilde{W}_{t+1} = \tilde{W}_t - \frac{\eta}{B} \sum_{i=1}^{B} \boldsymbol{a}_{\xi_i^t} \boldsymbol{a}_{\xi_i^t}^T \tilde{W}_t = \left( I - \frac{\eta}{B} \sum_{i=1}^{B} H_{\xi_i^t} \right) \tilde{W}_t. \tag{12}$$

Next, we define the stability of the above dynamics.

**Definition 5.** For any $k \in \mathbb{N}^*$, we say $W^*$ is $k$-**th order linearly stable** for SGD with learning rate $\eta$ and batch size $B$, if there exists a constant $C$ (which may depend on $k$) that satisfies

$$\left\| \mathbb{E}\tilde{W}_t^{\otimes k} \right\|_F \leq C \left\| \mathbb{E}\tilde{W}_0^{\otimes k} \right\|_F,$$

for any $\tilde{W}_t$, $t \in \mathbb{N}^*$, given by the dynamics (12) from any initialization distribution of $\tilde{W}_0$.

$\{\tilde{W}_t\}$ is the trajectory of the linearized dynamics. It characterizes the performance of SGD around the minimum $W^*$, but does not equal to the original trajectory of SGD $\{W_t\}$. In practice, quadratic approximation of the loss function works in a neighborhood around the minimum. Hence, linear stability characterizes the local behavior of the iterators around the minimum. The $\tilde{W}_0$ in Definition 5 is the starting point of the linearized dynamics. It can be a point in the trajectory of the real dynamics, but it should not be understood as the initialization of the real dynamics ($W_0$). Since $\tilde{W}_t$ follows a linear dynamics, the $\tilde{W}_0$ here does not need to be close to $W^*$. Without linear stability of $\{\tilde{W}_t\}$, $\{W_t\}$ can never converge to the minimum. It is possible that $\{W_t\}$ oscillates around the minimum at a distance out of the reach of local quadratic approximation. However, empirical observations in [11] show that SGD usually converges to and oscillates in a region with good quadratic approximation. Hence, linear stability plays an important role in practice.

In [30], the following condition on the linear stability for $k = 2$ is provided.

**Proposition 1.** (**Theorem 1 in [30]**) *The global minimum $W^*$ is $2^{nd}$-order linearly stable for SGD with learning rate $\eta$ and batch size $B$ if*

$$\lambda_{\max}\left\{ (I - \eta H)^2 + \frac{\eta^2(n-B)}{B(n-1)}\Sigma \right\} \leq 1, \tag{13}$$

*where $H = \frac{1}{n}\sum\limits_{i=1}^{n} H_i$, and $\Sigma = \frac{1}{n}\sum\limits_{i=1}^{n} H_i^2 - H^2$.*

Shown by the proposition, the spectra of $H$ and $\Sigma$ influence the linear stability of SGD. Thus, in [30] the biggest eigenvalue of $H$ and $\Sigma$ are named sharpness and non-uniformity, respectively. The condition (13) is then be relaxed into a condition of sharpness and non-uniformity. Similar analysis was also conducted in [9] for linear least squares problems.

**Remark 3.** *The definition of stability in [30] concerns $\mathbb{E}\|\tilde{W}_t\|^2$, which is slightly different from the definition above when $k = 2$. However, since $\mathbb{E}\|\tilde{W}_t\|^2$ and $\|\mathbb{E}\tilde{W}_t^{\otimes 2}\|_F$ are equivalent, i.e. $\mathbb{E}\|\tilde{W}_t\|^2/\sqrt{w} \leq \|\mathbb{E}\tilde{W}_t^{\otimes 2}\|_F \leq \mathbb{E}\|\tilde{W}_t\|^2$, the two ways to define linear stability are equivalent.*

In the current work, we analyze the dynamics of $\mathbb{E}\tilde{W}_t^{\otimes k}$, which is a closed linear dynamics. As a comparison, the dynamics of $\mathbb{E}\|\tilde{W}_t\|^2$ is not closed, i.e. $\mathbb{E}\|\tilde{W}_{t+1}\|^2$ is not totally determined by $\mathbb{E}\|\tilde{W}_t\|^2$. By (12) we have

$$\tilde{W}_{t+1}^{\otimes k} = \left( I - \frac{\eta}{B}\sum_{i=1}^{B} H_{\xi_i^t} \right)^{\otimes k} \tilde{W}_t^{\otimes k}. \tag{14}$$

Let $\mathcal{I}_B$ be the set of all subsets of $\{1, 2, ..., n\}$ with cardinality $B$, and $\mathfrak{I} = \{i_1, ..., i_B\} \in \mathcal{I}_B$ be a batch. Note that $\xi^t$ is independent with $\tilde{W}_t$. Taking expectation for (14) gives

$$\mathbb{E}\tilde{W}_{t+1}^{\otimes k} = \mathbb{E}\left( I - \frac{\eta}{B}\sum_{i=1}^{B} H_{\xi_i^t} \right)^{\otimes k} \mathbb{E}\tilde{W}_t^{\otimes k} = \frac{1}{\binom{n}{B}}\sum_{\mathfrak{I} \in \mathcal{I}}\left( I - \frac{\eta}{B}\sum_{j=1}^{B} H_{i_j} \right)^{\otimes k} \mathbb{E}\tilde{W}_t^{\otimes k} \tag{15}$$

Denote

$$T_{\mathfrak{I},k}^{\eta,B} := \left( I - \frac{\eta}{B}\sum_{j=1}^{B} H_{i_j} \right)^{\otimes k} \tag{16}$$

for each batch $\mathfrak{I}$, and $T_k^{\eta,B} = 1/\binom{n}{B}\sum_{\mathfrak{I}} T_{\mathfrak{I},k}^{\eta,B}$ be the expectation of $T_{\mathfrak{I}}^{\eta,B}$ over the choice of batches. Then we have $\mathbb{E}\tilde{W}_{t+1}^{\otimes k} = T_k^{\eta,B}\mathbb{E}\tilde{W}_t^{\otimes k}$.

On the other hand, if $\mathbb{E}\tilde{W}_t^{\otimes k}$ is understood as a tensor in $\mathbb{R}^{w \times w \times \cdots \times w}$, then it is always a symmetric tensor (any permutation of the index does not change the entry value). Hence, it has the decomposition $\mathbb{E}\tilde{W}_t^{\otimes k} = \sum_{i=1}^r \lambda_i \boldsymbol{v}_i^{\otimes k}$, with $\lambda_i \in \mathbb{R}$ and $v_i \in \mathbb{R}^w$ [8]. Let $\mathcal{M}_k$ be the set of symmetric tensors in $\mathbb{R}^{w \times w \times \cdots \times w}$, and $\mathcal{M}_k^+ = \{A \in \mathcal{M} : A = \sum_{i=1}^r \lambda_i v_i^{\otimes k} \text{ and } \lambda_i \geq 0 \text{ for any } i = 1, ..., r\}$ be the set of "positive semidefinite" symmetric tensors. We provide the following conditions for the linear stability of SGD.

**Theorem 1.** *For any $k \in \mathbb{N}^*$, the global minimum $W^*$ is $k^{th}$ order linearly stable for SGD with learning rate $\eta$ and batch size $B$, if and only if*

$$\|T_k^{\eta,B} A\|_F \leq \|A\|_F \tag{17}$$

*holds for any $A \in \mathcal{M}_k^+$ if $k$ is an even number, or for any $A \in \mathcal{M}_k$ is $k$ is an odd number.*

All proofs are provided in the appendix. As a corollary, when $k = 2$, we can write down another sufficient condition for linear stability than Proposition 1. See Corollary 3 in the appendix.

Recall that we let $\boldsymbol{a}_i = \nabla_W f(\mathbf{x}_i, W^*)$ and $H_i = \boldsymbol{a}_i \boldsymbol{a}_i^T$. The linear stability conditions in Theorem 1 imply the following bound on the moments of $\boldsymbol{a}_i$. Specifically, we have

**Theorem 2.** *If a global minimum $W^*$ is $k^{th}$ order linearly stable for SGD with learning rate $\eta$ and batch size $B$, then, for any $j \in \{1, 2, ..., p\}$, we have*

$$\frac{1}{n} \sum_{i=1}^n \boldsymbol{a}_{i,j}^{2k} \leq \frac{2^k B^{k-1}}{\eta^k}, \tag{18}$$

*where $\boldsymbol{a}_{i,j}$ means the $j^{th}$ entry of $\boldsymbol{a}_i$.*

By Theorem 2, if a global minimum is stable for SGD with some order, then the gradient moment of this order is controlled. Summing $j$ from 1 to $w$, we have $\frac{1}{n} \sum_{i=1}^n \|\nabla_W f(\mathbf{x}_i, W)\|_{2k}^{2k} \leq \frac{2w(2B)^{k-1}}{\eta^k}$. For $k = 1$ and 2, this gives control to the flatness and non-uniformity of the minimum, respectively. For general $k$, applying the Hölder inequality on (18), we have the following corollary which bounds the mean $2k$-norm of the gradients at the training data.

**Corollary 1.** *If a global minimum $W^*$ is $k^{th}$ order linearly stable for SGD with learning rate $\eta$ and batch size $B$, then*

$$\frac{1}{n} \sum_{i=1}^n \|\nabla_W f(\mathbf{x}_i, W^*)\|_{2k} = \frac{1}{n} \sum_{i=1}^n \|\boldsymbol{a}_i\|_{2k} \leq \left(\frac{w}{B}\right)^{\frac{1}{2k}} \sqrt{\frac{2B}{\eta}}. \tag{19}$$

## 4   The Sobolev regularization effect

In this section, we build connection between $\nabla_W f(\mathbf{x}, W)$ and $\nabla_\mathbf{x} f(\mathbf{x}, W)$ using the multiplication of $W_1$ and $\mathbf{x}$. For general parameterized model, it is hard to build connection between the two gradients. However, this is possible for neural network models, in which the input variable is multiplied with a set of parameters (the first-layer parameter) before any non-linear operation. By this connection, at an interpolation solution that is stable for SGD, we can turn the moments bounds derived in the previous section into the bounds on $\nabla_\mathbf{x} f(\mathbf{x}, W)$. For different $k$, the moment bound on $\nabla_W f(\mathbf{x}, W)$ controls the Sobolev seminorms of $f(\cdot, W)$ at the training data with different index $p$. When $k = 1$, this becomes a bound for the sum of gradient square at the training data, which is used in the literature as a regularization term to regularize the Lipschitz constant of the model function. This explains the favorable generalization performance of flat minima.

Recall $f(\mathbf{x}, W) = \tilde{f}(W_1 \mathbf{x}, W_2)$, and we can express $\nabla_\mathbf{x} \tilde{f}$ and $\nabla_{W_1} \tilde{f}$ as (1). Then, we easily have the following proposition.

**Proposition 2.** *Consider the model $\tilde{f}(W_1 \mathbf{x}, W_2)$. For any $k \in \mathbb{N}^*$, any $W = (W_1, W_2)$ and $\mathbf{x} \neq 0$,*

$$\|\nabla_\mathbf{x} \tilde{f}(W_1 \mathbf{x}, W_2)\|_{2k}^{2k} \leq \frac{\|W_1^T\|_{2k}^{2k}}{\|\mathbf{x}\|_{2k}^{2k}} \|\nabla_W \tilde{f}(W_1 \mathbf{x}, W_2)\|_{2k}^{2k}, \tag{20}$$

As a corollary, we have the following bound for the Sobolev seminorm using $\nabla_W \tilde{f}$.

**Corollary 2.** *Let $f(\mathbf{x}, W) = \tilde{f}(W_1\mathbf{x}, W_2)$, and $\{\mathbf{x}_i\} = \{\mathbf{x}_1, \mathbf{x}_2, ..., \mathbf{x}_n\}$. Let $|f(\cdot, W)|_{1,2k,\{\mathbf{x}_i\}}$ be the Sobolev seminorm of $f(\cdot, W)$ on $\{\mathbf{x}_i\}$ as defined at the beginning of Section 2. Then,*

$$|f(\cdot, W)|_{1,2k,\{\mathbf{x}_i\}} \leq \frac{\|W_1^T\|_{2k}}{\min_{1 \leq i \leq n} \|\mathbf{x}_i\|_{2k}} \left( \frac{1}{n} \sum_{i=1}^{n} \|\nabla_W f(\mathbf{x}_i, W)\|_{2k}^{2k} \right)^{\frac{1}{2k}}. \tag{21}$$

Combining Corollary 2 with the stability condition in Theorem 2, we have the following control for the Sobolev seminorm of the model function at interpolation solutions that are stable for SGD.

**Theorem 3.** *If a global minimum $W^*$ which interpolates the training data is $k^{th}$ order linearly stable for SGD with learning rate $\eta$ and batch size $B$, and $W^* = (W_1^*, W_2^*)$. Then,*

$$|f(\cdot, W^*)|_{1,2k,\{\mathbf{x}_i\}} \leq \frac{\|(W_1^*)^T\|_{2k}}{\min_{1 \leq i \leq n} \|\mathbf{x}_i\|_{2k}} \left( \frac{w}{B} \right)^{\frac{1}{2k}} \sqrt{\frac{2B}{\eta}}. \tag{22}$$

By Corollary 2 and Theorem 3, as long as the data are not very small (in practical problems, the input data are usually normalized), and the input-layer parameters are not very large, the Sobolev seminorms of the model function (evaluated at the training data) is regularized by the linear stability of SGD. The regularization effect on Sobolev seminorm gets stronger for bigger learning rate and smaller batch size. When $k$ is big, the dependence of the bound with $p$ is negligible.

If the solution $W^*$ satisfies the local smoothness condition defined in Definition 1, the control on $\nabla_\mathbf{x} \tilde{f}(W_1^*\mathbf{x}, W_2^*)$ can be extended to a neighborhood around the training data. Then, we will be able to control the "population" Sobolev seminorms. First, around a certain training data, we have the following estimate.

**Proposition 3.** *Assume the model $f(\mathbf{x}, W)$ satisfies $(C, \delta_{approx}, k)$-local smoothness condition for some $C, \delta_{approx} > 0$ and $k \in \mathbb{N}^*$, at $W^*$ and $\mathbf{x}_*$. Then, for any $\mathbf{x}$ that satisfies $\|\mathbf{x} - \mathbf{x}_*\|_2 \leq \delta_{approx}\|\mathbf{x}_*\|_2 / \|W_1^*\|_2$, we have*

$$\|\nabla_\mathbf{x} f(\mathbf{x}, W^*)\|_{2k} \leq \frac{C\|(W_1^*)^T\|_{2k}}{\|\mathbf{x}_*\|_{2k}} \left( \|\nabla_W f(\mathbf{x}_*, W^*)\|_{2k} + 1 \right). \tag{23}$$

Proposition 3 turns the local smoothness condition in the parameter space into a local smoothness condition in the data space. This is made possible by the multiplicative structure of the network's input layer. Specifically, since in the first layer $W_1$ and $\mathbf{x}$ are multiplied together, a perturbation of $\mathbf{x}$ can be turned to a perturbation of $W_1$ without changing the value of their product. See the proof of the proposition in Appendix B.2 for more detail.

By the results above, we can obtain the following theorem which estimates the Sobolev seminorm of the model function on a neighborhood of the training data.

**Theorem 4.** *Let $W^* = (W_1^*, W_2^*)$ be an interpolation solution that is $k^{th}$ order linearly stable for SGD with learning rate $\eta$ and batch size $B$. Let $\{\mathbf{x}_1, ..., \mathbf{x}_n\}$ be the training data. Suppose the model $f(\mathbf{x}, W)$ satisfies $(C, \delta_{approx}, k)$-local smoothness condition at $W^*$ and $x_i$ for any $i = 1, 2, ..., n$. Consider the set $\mathcal{X}_\delta := \bigcup_{i=1}^{n} \{\mathbf{x}: \|\mathbf{x} - \mathbf{x}_i\| \leq \delta\}$ with $\delta \leq \delta_{approx}\|\mathbf{x}_i\|_2 / \|W_1^*\|_2$. Assume $\{\mathbf{x}_1, ..., \mathbf{x}_n\}$ satisfies $(\delta, K)$-scattered condition as defined in Definition 3. Then, we have*

$$|f(\mathbf{x}, W^*)|_{1,2k,\mathcal{X}_\delta} \leq \frac{(KV_{\mathcal{X}_\delta})^{\frac{1}{2k}} 2C\|(W_1^*)^T\|_{2k}}{\min_i \|\mathbf{x}_i\|_{2k}} \left( \left( \frac{w}{B} \right)^{\frac{1}{2k}} \sqrt{\frac{2B}{\eta}} + 1 \right), \tag{24}$$

*where $V_{\mathcal{X}_\delta}$ is the Lebesgue volume of $\mathcal{X}_\delta$.*

By Theorem 4, the model function found by SGD is smooth in a neighborhood of the training data, given that the landscape of the model in the parameter space is smooth. When $k = 1$, the results implies that flatness regularizes $|\nabla_\mathbf{x} f(\mathbf{x}, W^*)|_{1,2,\mathcal{X}_\delta}$, which is stronger than the gradient square at training data.

**Discussion on the tightness of the bounds** The bounds in this section depend on the norm of $W_1^*$. This norm may be big, especially when homogeneous activation functions, such as ReLU, is used. In

| iterations | 0 | 100 | 1000 | 10000 | 20000 | 50000 |
|---|---|---|---|---|---|---|
| $\|W_1\|_2$ | 1.032 | 1.034 | 1.218 | 2.011 | 2.257 | 2.270 |

Table 1: The average $l_2$ norm of the first-layer parameter matrix of a fully-connected neural network trained on FashionMNIST dataset.

the ReLU case, for example, we can simply scale the first layer parameters to be arbitrarily large and second layer small while not changing the model function.

However, in practice SGD does not find these solutions with unbalanced layers because the homogeneity of ReLU induces an invariant in parameters. Specifically, let $W_l$ and $W_{l-1}$ be the parameter matrices in the $l^{\text{th}}$ and $(l-1)^{\text{th}}$ layers, then $W_{l-1}^T W_{l-1} - W_l W_l^T$ does not change significantly throughout the training (it is actually fixed in the zero learning rate limit). This invariant keeps layers balanced and prevents the first layer parameters from getting big alone.

On the other hand, we conduct numerical experiments to check the norm of $W_1$ during training. Table 1 shows the average $l_2$-norm of $W_1$ for the same experiments in part (a) of Figure 1. The model is a fully-connected neural network, and the dataset is FashionMNIST. Results show that the norm of $W_1$ increases at the beginning, but becomes stable afterwards. It will not keep increasing during the training.

## 5 Generalization error and adversarial robustness

Regularizing the smoothness of the model function can improve generalization performance and adversarial robustness. This is confirmed in many practical studies [24, 29, 26]. Intuitively, a function with small gradient changes slowly as the input data changes, hence when the test data is close to one of the training data, the prediction error on the test data will be small. In this section, going along this intuition, we provide theoretical analysis of the generalization performance and adversarial robustness based on the Sobolev regularization effect derived in the previous sections. Still consider the model $f(\mathbf{x}, W) = \tilde{f}(W_1\mathbf{x}, W_2)$, and an interpolation solution $W^* = (W_1^*, W_2^*)$ found by SGD. First, based on Proposition 3 and Theorem 2, we show the following theorem, which is similar to Theorem 4 but estimates the maximum value of $\|\nabla_{\mathbf{x}} f(\mathbf{x}, W^*)\|$ in $\mathcal{X}_\delta$.

**Theorem 5.** *Let $W* = (W_1^*, W_2^*)$ be an interpolation solution that is $k^{th}$ order linearly stable for SGD with learning rate $\eta$ and batch size $B$. Let $\{\mathbf{x}_1, ..., \mathbf{x}_n\}$ be the training data. Suppose the model $f(\mathbf{x}, W)$ satisfies $(C, \delta_{approx}, k)$-local smoothness condition at $W^*$ and $\mathbf{x}_i$ for any $1 = 1, 2, ..., n$. Recall the definition of $\mathcal{X}_\delta$ in Theorem 4. Then, for any $\mathbf{x} \in \mathcal{X}_\delta$, we have*

$$\|\nabla_{\mathbf{x}} f(\mathbf{x}, W^*)\|_{2k} \leq \frac{C\|(W_1^*)^T\|_{2k}}{\min_i \|\mathbf{x}_i\|_{2k}} \left( \left(\frac{2nw}{B}\right)^{\frac{1}{2k}} \sqrt{\frac{2B}{\eta}} + 1 \right). \tag{25}$$

With the result above, we can bound the generalization error if most test data are close to the training data, i.e. the data distribution $\mu$ satisfies a covered condition. This happens for machine learning problems with sufficient training data, especially for those where the training data lie approximately on a union of some low-dimensional surfaces. This is common in practical problems [1].

**Theorem 6.** *Suppose parameterized model $f(\mathbf{x}, W) = \tilde{f}(W_1\mathbf{x}, W_2)$ is used to solve a supervised learning problem with data distribution $\mu$. Let $f^*$ be the target function. Assume $\mu$ satisfies $(n, \delta, \varepsilon_1, \varepsilon_2)$-covered condition. Assume with probability no less than $1 - \varepsilon_1$ SGD with learning rate $\eta$ and batch size $B$ can find an interpolation solution $W^*$ which is $k^{th}$ order linearly stable, and $(C, \delta_{approx}, k)$-local smoothness condition is satisfied at $W^*$ and $\mathbf{x}_1, ..., \mathbf{x}_n$ with $\delta_{approx} \geq \delta\|W_1^*\|_2 / \min_i\{\|\mathbf{x}_i\|_2\}$. Further assume that both $|f(\cdot, W^*)|$ and $|f^*(\cdot)|$ are upper bounded by $M_1$, and $\|\nabla_{\mathbf{x}} f^*(\cdot)\|_2$ is upper bounded by $M_2$, for some constants $M_1$ and $M_2$. Then, with probability $1 - 2\varepsilon_1$ over the sampling of training data $\{\mathbf{x}_i\}_{i=1}^n$, we have*

$$\mathbb{E}_{\mathbf{x}\sim\mu}\|f(\mathbf{x}, W^*) - f^*(\mathbf{x})\|_2^2 \leq \frac{2dC^2\|(W_1^*)^T\|_{2k}^2}{\min_i \|\mathbf{x}_i\|_{2k}^2} \left( \left(\frac{2nw}{B}\right)^{\frac{1}{2k}} \sqrt{\frac{2B}{\eta}} + 1 \right)^2 \delta^2 + 2M_2^2\delta^2 + 4M_1^2\varepsilon_2.$$

$$\tag{26}$$

The bound (26) tends to 0 as $n \to \infty$ as long as $\delta$ decays faster than $n^{-\frac{1}{2k}}$. From a geometric perspective, this happens when the dimension of the support of $\mu$ is less than $k$. And the lower the dimension, the faster the decay. It is possible to get rid of the $n$ dependency in the bound when $k$ is sufficiently large. We may use the estimate of Sobolev functions with scattered zeros, e.g. Theorem 4.1 in [2]. We leave the analysis to future work.

**Adversarial robustness**   Neural network models suffer from adversarial examples [28], because the model function changes very fast in some directions so the function value becomes very different after a small perturbation. However, the results in Theorem 5 directly imply the adversarial robustness of the model at the training data. Hence, flatness, as well as high-order linear stability conditions, also imply adversarial robustness. Specifically, we have the following theorem.

**Theorem 7.** *Let $W* = (W_1^*, W_2^*)$ be an interpolation solution that is $k^{th}$ order linearly stable for SGD with learning rate $\eta$ and batch size $B$. Let $\{\mathbf{x}_1, ..., \mathbf{x}_n\}$ be the training data. Suppose the model $f(\mathbf{x}, W)$ satisfies $(C, \delta_{approx}, k)$-local smoothness condition at $W^*$ and $\mathbf{x}_i$ for any $1 = 1, 2, ..., n$. Then, for any $\mathbf{x}$ that satisfies $\|\mathbf{x} - \mathbf{x}_i\|_2 \leq \delta$ for some $i \in \{1, ..., n\}$ and $\delta \leq \delta_{approx} \min_i \|\mathbf{x}_i\|_2 / \|W_1^*\|_2$, we have*

$$|f(\mathbf{x}, W^*) - f^*(\mathbf{x}_i, W^*)| \leq \frac{C\sqrt{d}\|(W_1^*)^T\|_{2k}}{\min_i \|\mathbf{x}_i\|_{2k}} \left( \left(\frac{2nw}{B}\right)^{\frac{1}{2k}} \sqrt{\frac{2B}{\eta}} + 1 \right) \delta. \qquad (27)$$

## 6   Conclusion

In this paper, we connect the linear stability theory of SGD with the generalization performance and adversarial robustness of the neural network model. As a corollary, we provide theoretical insights of why flat minimum generalizes better. To achieve the goal, we explore the multiplicative structure of the neural network's input layer, and build connection between the model's gradient with respect to the parameters and the gradient with respect to the input data. We show that as long as the landscape on the parameter space is mild, the landscape of the model function with respect to the input data is also mild, hence the flatness (as well as higher order linear stability conditions) has the effect of Sobolev regularization. Our study reveals the significance of the multiplication structure between data (or features in intermediate layers) and parameters. It is an important source of implicit regularization of neural networks and deserves further exploration.

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
