# A  Proofs for Section 3

## A.1  Proof of Theorem 1

In the proof, we ignore the superscripts $\eta$ and $B$. We first show the sufficiency. Assume (17) holds. For any distribution of $\tilde{W}_0$, obviously we have $\mathbb{E}\tilde{W}_0^{\otimes k} \in \mathcal{M}_k$. Hence, if $k$ is odd, linear stability comes directly from (17). If $k$ is even, for any vector $\boldsymbol{v} \in \mathbb{R}^w$, we have

$$\mathbb{E}\tilde{W}_0^{\otimes k} \cdot \boldsymbol{v}^{\otimes k} = \mathbb{E}(\tilde{W}_0^T \boldsymbol{v})^k \geq 0,$$

which means $\mathbb{E}\tilde{W}_0^{\otimes k} \in \mathcal{M}_k^+$. Thus, the $k^{\text{th}}$-order linear stability also holds for this distribution of $\tilde{W}_0$.

Next, we show the necessity. Let $A \in \mathcal{M}_k$, then $A$ has the following decomposition

$$A = \sum_{i=1}^{r} \lambda_i \boldsymbol{v}_i^{\otimes k}$$

where $r \in \mathbb{N}^*$, $\lambda_i$ are real numbers and $\boldsymbol{v}_i \in \mathbb{R}^w$. Then,

$$T_k A = \sum_{i=1}^{r} \lambda_i T_k(\boldsymbol{v}_i^{\otimes k}) = \frac{1}{\binom{n}{B}} \sum_{\mathfrak{J} \in \mathcal{I}} \sum_{i=1}^{r} \lambda_i \left( \left( I - \frac{\eta}{B} \sum_{j=1}^{B} H_{i_j} \right) \boldsymbol{v}_i \right)^{\otimes k}. \tag{28}$$

Hence, $T_k A$ is still symmetric, i.e. $T_k A \in \mathcal{M}_k$. Therefore, $T_k$ induces a linear transform from $\mathcal{M}_k$ to $\mathcal{M}_k$. Let $\mathcal{T}_k$ be this linear transform. Since $H_i$ is symmetric for all $i = 1, 2, ..., n$, if we understand $T_{\mathfrak{J}}$ as a matrix in $\mathbb{R}^{w^k \times w^k}$, then $T_{\mathfrak{J}}$ is symmetric for any batch $\mathfrak{J}$. Therefore, $T_k$ is symmetric, which means $\mathcal{T}_k$ is also a symmetric linear transform. Then, we can easily show the following lemma by eigen-decomposition of $\mathcal{T}_k$:

**Lemma 1.** *For any $A \in \mathcal{M}_k$ and $A \neq 0$, if $\|\mathcal{T}_k A\|_F > \|A\|_F$, then $\lim_{m \to \infty} \|(\mathcal{T}_k)^m A\|_F = \infty$.*

The lemma is proven in Section D. With the lemma, the necessity follows by showing that we can find a distribution of $\tilde{W}_0$ such that $\mathbb{E}\tilde{W}_0^{\otimes k} = A$ for any $A \in \mathcal{M}_k^+$ if $k$ is even and $A \in \mathcal{M}_k$ if $k$ is odd. First consider an even $k$. For any $A \in \mathcal{M}_k^+$, we have the decomposition

$$A = \sum_{i=1}^{r} \lambda_i \boldsymbol{v}_i^{\otimes k}, \tag{29}$$

where $\lambda_i \geq 0$ for $i = 1, 2, ..., r$. Let the probability distribution of $\tilde{W}_0$ be given by the density function

$$p(W) := \sum_{i=1}^{r} \frac{\lambda_i}{\sum_{j=1}^{r} \lambda_j} \delta \left( W - (\sum_{j=1}^{r} \lambda_j)^{\frac{1}{k}} \boldsymbol{v}_i \right).$$

Then, we have $\mathbb{E}\tilde{W}_0^{\otimes k} = A$. Next, if $k$ is odd, for any $A \in \mathcal{M}_k$, we still have decomposition (29), but some $\lambda_i$ may be negative. However, since now $k$ is an odd number, we can write the decomposition as

$$A = \sum_{i=1}^{r} |\lambda_i| (\text{sign}(\lambda_i)\boldsymbol{v}_i)^{\otimes k}.$$

Then, a similar construction as in the even case completes the proof.

## A.2  Corollary 3 and the proof

**Corollary 3.** *The global minimum $W^*$ is $2^{nd}$-order linearly stable for SGD with learning rate $\eta$ and batch size $B$ if*

$$\max \left| \lambda \left( (I - \eta H)^{\otimes 2} + \frac{(n-B)}{B(n-1)} \frac{\eta^2}{n} \sum_{i=1}^{n} (H_i^{\otimes 2} - H^{\otimes 2}) \right) \right| \leq 1 \tag{30}$$

*Proof:*

When $k = 2$ we have

$$T_2^{\eta,B} = \mathbb{E}_{\mathfrak{J}}\left(I - \frac{\eta}{B}\sum_{j=1}^{B}H_{i_j}\right)^{\otimes 2}$$

$$= \mathbb{E}_{\mathfrak{J}}\left(I^{\otimes 2} - \frac{\eta}{B}\sum_{j=1}^{B}(I \otimes H_{i_j} + H_{i_j} \otimes I) + \frac{\eta^2}{B^2}\sum_{j_1,j_2=1}^{B}H_{i_{j_1}} \otimes H_{i_{j_2}}\right)$$

$$= I^{\otimes 2} - \eta(I \otimes H + H \otimes I) + \frac{\eta^2}{B^2}\sum_{j_1,j_2=1}^{B}\mathbb{E}_{\mathfrak{J}}(H_{i_{j_1}} \otimes H_{i_{j_2}}). \tag{31}$$

For each $i \in \{1, 2, ..., n\}$, $H_i$ appears in $\binom{n-1}{B-1}$ batches, and for each $(i,j)$, $i, j \in \{1, 2, ..., n\}$, $H_i$ and $H_j$ appears in $\binom{n-2}{B-2}$ batches simultaneously. Hence,

$$\mathbb{E}_{\mathfrak{J}}\sum_{j=1}^{B}H_{i_j} \otimes H_{i_j} = \sum_{i=1}^{n}\frac{\binom{n-1}{B-1}}{\binom{n}{B}}H_i \otimes H_i = \frac{B}{n}\sum_{i=1}^{n}H_i \otimes H_i$$

$$\mathbb{E}_{\mathfrak{J}}\sum_{j_1 \neq j_2}H_{i_{j_1}} \otimes H_{i_{j_2}} = \sum_{i \neq j}\frac{\binom{n-2}{B-2}}{\binom{n}{B}}H_i \otimes H_j = \frac{B(B-1)}{n(n-1)}\sum_{i \neq j}H_i \otimes H_j.$$

Therefore, we have

$$\frac{\eta^2}{B^2}\sum_{j_1,j_2=1}^{B}\mathbb{E}_{\mathfrak{J}}(H_{i_{j_1}} \otimes H_{i_{j_2}}) = \frac{\eta^2}{nB}\sum_{i=1}^{n}H_i \otimes H_i + \frac{\eta^2(B-1)}{Bn(n-1)}\sum_{i \neq j}H_i \otimes H_j$$

$$= \frac{\eta^2(B-1)}{Bn(n-1)}\sum_{i,j=1}^{n}H_i \otimes H_j + \left(\frac{\eta^2}{nB} - \frac{\eta^2(B-1)}{Bn(n-1)}\right)\sum_{i=1}^{n}H_i \otimes H_i$$

$$= \frac{\eta^2 n(B-1)}{B(n-1)}H \otimes H + \frac{\eta^2(n-B)}{Bn(n-1)}\sum_{i=1}^{n}H_i \otimes H_i$$

$$= \eta^2 H \otimes H + \frac{(n-B)}{B(n-1)}\frac{\eta^2}{n}\sum_{i=1}^{n}(H_i^{\otimes 2} - H^{\otimes 2}). \tag{32}$$

Plug (32) into (31), we have

$$T_2^{\eta,B} = (I - \eta H)^{\otimes 2} + \frac{(n-B)}{B(n-1)}\frac{\eta^2}{n}\sum_{i=1}^{n}(H_i^{\otimes 2} - H^{\otimes 2}). \tag{33}$$

Then, the result is a direct application of Theorem 1.

## A.3 Proof of Theorem 2

By Theorem 1, for any $A \in \mathcal{M}_k^+$ we have $\|T_k^{\eta,B}A\|_F \leq \|A\|_F$. For any $j \in \{1, 2, ..., w\}$, let $e_j \in \mathbb{R}^w$ be the $j^{\text{th}}$ unit coordinate vector, and let $A_j = e_j^{\otimes k}$. Then $\|A\|_F = 1$. On the other hand,

$$(T_k^{\eta,B}A_j)_{j,j,...,j} = \frac{1}{\binom{n}{B}}\sum_{\mathfrak{J}}(T_{\mathfrak{J},k}^{\eta,B}A_j)_{j,j,...,j}$$

$$= \frac{1}{\binom{n}{B}}\sum_{\mathfrak{J}}\left(1 - \frac{\eta}{B}\sum_{k=1}^{B}a_{i_k,j}^2\right)^k$$

Hence,

$$\left|\frac{1}{\binom{n}{B}}\sum_{\mathfrak{J}}\left(1 - \frac{\eta}{B}\sum_{k=1}^{B}a_{i_k,j}^2\right)^k\right| \leq \|T_k^{\eta,B}A_j\|_F \leq 1. \tag{34}$$

Next, we will use the following lemma, whose proof is also provided in the appendix.

**Lemma 2.** *For any $t \geq 0$ and $k \in \mathbb{N}^*$, we have*

$$t^k \leq 2^{k-1}((t-1)^k + 1).$$

For any batch $\mathfrak{I} = \{i_1, i_2, ..., i_B\}$, let $t = \eta/B \sum_{k=1}^B \boldsymbol{a}_{i_k,j}^2$, we obtain

$$\left( \frac{\eta}{B} \sum_{k=1}^B \boldsymbol{a}_{i_k,j}^2 \right)^k \leq 2^{k-1} \left( \left( \frac{\eta}{B} \sum_{k=1}^B \boldsymbol{a}_{i_k,j}^2 - 1 \right)^k + 1 \right).$$

Together with

$$\frac{\eta^k}{B^k} \sum_{k=1}^B \boldsymbol{a}_{i_k,j}^{2k} \leq \left( \frac{\eta}{B} \sum_{k=1}^B \boldsymbol{a}_{i_k,j}^2 \right)^k,$$

we have

$$\frac{1}{B} \sum_{k=1}^B \boldsymbol{a}_{i_k,j}^{2k} \leq \frac{2^{k-1} B^{k-1}}{\eta^k} \left( \left( \frac{\eta}{B} \sum_{k=1}^B \boldsymbol{a}_{i_k,j}^2 - 1 \right)^k + 1 \right). \tag{35}$$

Taking expectation over batches, by (34) we have

$$\frac{1}{n} \sum_{i=1}^n \boldsymbol{a}_{i,j}^{2k} \leq \frac{(2B)^{k-1}}{\eta^k} \left( \frac{1}{\binom{n}{B}} \sum_{\mathfrak{I}} \left( \frac{\eta}{B} \sum_{k=1}^B \boldsymbol{a}_{i_k,j}^2 - 1 \right)^k + 1 \right) \leq \frac{2(2B)^{k-1}}{\eta^k}. \tag{36}$$

# B Proofs for Section 4

## B.1 Proof of Proposition 2

In this proof, $\| \cdot \|_{2k}$ always means the vector or matrix $2k$-norm, not the function norm. Then, we have

$$\|\nabla_{\mathbf{x}} \tilde{f}(W_1\mathbf{x}, W_2)\|_{2k}^{2k} = \left\| W_1^T \frac{\partial \tilde{f}(W_1\mathbf{x}, W_2)}{\partial(W_1\mathbf{x})} \right\|_{2k}^{2k} \leq \|W_1^T\|_{2k}^{2k} \left\| \frac{\partial \tilde{f}(W_1\mathbf{x}, W_2)}{\partial(W_1\mathbf{x})} \right\|_{2k}^{2k}.$$

On the other hand,

$$\sum_{j=1}^m \sum_{l=1}^d [\nabla_{W_1} \tilde{f}(W_1\mathbf{x}, W_2)]_{jl}^{2k} = \left\| \frac{\partial \tilde{f}(W_1\mathbf{x}, W_2)}{\partial(W_1\mathbf{x})} \right\|_{2k}^{2k} \|\mathbf{x}\|_{2k}^{2k}.$$

Hence,

$$\sum_{j=1}^d [\nabla_{\mathbf{x}} \tilde{f}(W_1\mathbf{x}, W_2)]_j^{2k} \leq \frac{\|W_1^T\|_{2k}^{2k}}{\|\mathbf{x}\|_{2k}^{2k}} \sum_{j=1}^m \sum_{l=1}^d [\nabla_{W_1} \tilde{f}(W_1\mathbf{x}, W_2)]_{jl}^{2k}.$$

Since $W_1$ is a subset of $W$, obviously we have

$$\sum_{j=1}^m \sum_{l=1}^d [\nabla_{W_1} \tilde{f}(W_1\mathbf{x}, W_2)]_{jl}^{2k} \leq \|\nabla_W \tilde{f}(W_1\mathbf{x}, W_2)\|_{2k}^{2k},$$

which completes the proof.

## B.2 Proof of Proposition 3

Recall that $f(\mathbf{x}, W) = \tilde{f}(W_1\mathbf{x}, W_2)$. First, we find a $W_1$ such that $W_1\mathbf{x}_* = W_1^*\mathbf{x}$. Let $V = W_1 - W_1^*$, this is equivalent with solving the linear system

$$V\mathbf{x}_* = W_1^*(\mathbf{x} - \mathbf{x}_*)$$

for $V$. The linear system above is under-determined, hence solutions exist. We take the minimal norm solution

$$V = \frac{1}{\|\mathbf{x}_*\|_2^2}(W_1^*)^T(\mathbf{x} - \mathbf{x}_*)\mathbf{x}_*^T.$$

Especially, we have

$$\|W_1 - W_1^*\|_F = \|V\|_F = \frac{\|(W_1^*)^T(\mathbf{x} - \mathbf{x}_*)\|_2}{\|\mathbf{x}_*\|_2} \leq \frac{\|W_1^*\|_2}{\|\mathbf{x}_*\|_2}\|\mathbf{x} - \mathbf{x}_*\|_2 \leq \delta_{\text{approx}}.$$

Next, since $W_1\mathbf{x}_* = W_1^*\mathbf{x}$, we have $\tilde{f}(W_1^*\mathbf{x}, W_2^*) = \tilde{f}(W_1\mathbf{x}_*, W_2^*)$, and for gradient we have

$$\begin{aligned}
\|\nabla_{\mathbf{x}}\tilde{f}(W_1^*\mathbf{x}, W_2^*)\|_{2k} &\leq \frac{\|(W_1^*)^T\|_{2k}}{\|\mathbf{x}\|_{2k}}\|\nabla_{W_1}\tilde{f}(W_1^*\mathbf{x}, W_2^*)\|_{2k} \\
&= \frac{\|(W_1^*)^T\|_{2k}}{\|\mathbf{x}\|_{2k}}\left\|\frac{\partial\tilde{f}(W_1^*\mathbf{x}, W_2^*)}{\partial(W_1^*\mathbf{x})}\right\|_{2k}\|\mathbf{x}\|_{2k} \\
&= \frac{\|(W_1^*)^T\|_{2k}}{\|\mathbf{x}\|_{2k}}\left\|\frac{\partial\tilde{f}(W_1\mathbf{x}_*, W_2^*)}{\partial(W_1^*\mathbf{x})}\right\|_{2k}\|\mathbf{x}\|_{2k}\frac{\|\mathbf{x}_*\|_{2k}}{\|\mathbf{x}_*\|_{2k}} \\
&= \frac{\|(W_1^*)^T\|_{2k}}{\|\mathbf{x}_*\|_{2k}}\|\nabla_{W_1}\tilde{f}(W_1\mathbf{x}_*, W_2^*)\|_{2k}. \tag{37}
\end{aligned}$$

Let $W = (W_1, W_2^*)$, then $\|W - W^*\|_2 = \|W_1 - W_1^*\|_F \leq \delta_{\text{approx}}$. Hence, by (8) we have

$$\|\nabla_{W_1}\tilde{f}(W_1\mathbf{x}_*, W_2^*)\|_{2k} \leq \|\nabla_W\tilde{f}(W_1\mathbf{x}_*, W_2^*)\|_{2k} \leq C\left(\|\nabla_W\tilde{f}(W_1^*\mathbf{x}_*, W_2^*)\|_{2k} + 1\right).$$

This together with (37) completes the proof of (23).

### B.3 Proof of Theorem 4

Let $B(\mathbf{x}_i\delta)$ be the ball in $\mathbb{R}^d$ centered at $\mathbf{x}_i$ with radius $\delta$. Then, for any $\mathbf{x} \in B(\mathbf{x}_i, \delta)$, by Proposition 3 we have

$$\|\nabla_{\mathbf{x}}f(\mathbf{x}, W^*)\|_{2k} \leq \frac{C\|(W_1^*)^T\|_{2k}}{\|\mathbf{x}_i\|_{2k}}\left(\|\nabla_W f(\mathbf{x}_i, W^*)\|_{2k} + 1\right).$$

Hence,

$$\begin{aligned}
\int_{B(\mathbf{x}_i, \delta)}\|\nabla_{\mathbf{x}}f(\mathbf{x}, W^*)\|_{2k}^{2k}\mathbf{x} &\leq V_{B_\delta}\frac{C^{2k}\|(W_1^*)^T\|_{2k}^{2k}}{\|\mathbf{x}_i\|_{2k}^{2k}}\left(\|\nabla_W f(\mathbf{x}_i, W^*)\|_{2k} + 1\right)^{2k} \\
&\leq V_{B_\delta}\frac{2^{2k}C^{2k}\|(W_1^*)^T\|_{2k}^{2k}}{\|\mathbf{x}_i\|_{2k}^{2k}}\left(\|\nabla_W f(\mathbf{x}_i, W^*)\|_{2k}^{2k} + 1\right),
\end{aligned}$$

where $V_{B_\delta}$ is the volume of $B(\mathbf{x}_i, \delta)$, which does not depend on $\mathbf{x}_i$. Sum the above integral up for all training data, we have

$$\begin{aligned}
\int_{\mathcal{X}_\delta}\|\nabla_{\mathbf{x}}f(\mathbf{x}, W^*)\|_{2k}^{2k}\kappa(\mathbf{x})d\mathbf{x} &= \sum_{i=1}^n\int_{B(\mathbf{x}_i, \delta)}\|\nabla_{\mathbf{x}}f(\mathbf{x}, W^*)\|_{2k}^{2k}d\mathbf{x} \\
&\leq V_{B_\delta}\frac{2^{2k}C^{2k}\|(W_1^*)^T\|_{2k}^{2k}}{\min_i\|\mathbf{x}_i\|_{2k}^{2k}}\left(\sum_{i=1}^n\|\nabla_W f(\mathbf{x}_i, W^*)\|_{2k}^{2k} + n\right) \\
&\leq nV_{B_\delta}\frac{2^{2k}C^{2k}\|(W_1^*)^T\|_{2k}^{2k}}{\min_i\|\mathbf{x}_i\|_{2k}^{2k}}\left(\frac{2w(2B)^{k-1}}{\eta^k} + 1\right). \tag{38}
\end{aligned}$$

On the other hand,

$$\int_{\mathcal{X}_\delta}\|\nabla_{\mathbf{x}}f(\mathbf{x}, W^*)\|_{2k}^{2k}\kappa(\mathbf{x})d\mathbf{x} \geq \int_{\mathcal{X}_\delta}\|\nabla_{\mathbf{x}}f(\mathbf{x}, W^*)\|_{2k}^{2k}d\mathbf{x}.$$

Therefore,

$$\frac{1}{V_{\mathcal{X}_\delta}} \int_{\mathcal{X}_\delta} \|\nabla_{\mathbf{x}} f(\mathbf{x}, W^*)\|_{2k}^{2k} d\mathbf{x} \leq \frac{nV_{B_\delta}}{V_{\mathcal{X}_\delta}} \frac{2^{2k} C^{2k} \|(W_1^*)^T\|_{2k}^{2k}}{\min_i \|\mathbf{x}_i\|_{2k}^{2k}} \left( \frac{2w(2B^{k-1})}{\eta^k} + 1 \right)$$

$$= \frac{1}{V_{\mathcal{X}_\delta}} \int_{\mathcal{X}_\delta} \kappa(\mathbf{x}) d\mathbf{x} \frac{2^{2k} C^{2k} \|(W_1^*)^T\|_{2k}^{2k}}{\min_i \|\mathbf{x}_i\|_{2k}^{2k}} \left( \frac{2w(2B)^{k-1}}{\eta^k} + 1 \right)$$

$$\leq \frac{K 2^{2k} C^{2k} \|(W_1^*)^T\|_{2k}^{2k}}{\min_i \|\mathbf{x}_i\|_{2k}^{2k}} \left( \frac{2w(2B)^{k-1}}{\eta^k} + 1 \right). \tag{39}$$

Finally, we have

$$|f(\mathbf{x}, W^*)|_{1,2k,\mathcal{X}_\delta} \leq \frac{(KV_{\mathcal{X}_\delta})^{\frac{1}{2k}} 2C \|(W_1^*)^T\|_{2k}}{\min_i \|\mathbf{x}_i\|_{2k}} \left( \left( \frac{w}{B} \right)^{\frac{1}{2k}} \sqrt{\frac{2B}{\eta}} + 1 \right). \tag{40}$$

## C   Proofs for Section 5

### C.1   Proof of Theorem 5

Recall we let $\boldsymbol{a}_i = \nabla_W f(\mathbf{x}_i, W^*) \in \mathbb{R}^w$. By Theorem 2, we have

$$\frac{1}{n} \sum_{i=1}^n \boldsymbol{a}_{i,j}^{2k} \leq \frac{2^k B^{k-1}}{\eta^k},$$

for any $j = 1, 2, ..., w$. Therefore, for any $j$,

$$\max_{1 \leq i \leq n} \boldsymbol{a}_{ij}^{2k} \leq \frac{2^k B^{k-1} n}{\eta^k}.$$

Sum $j$ from 1 to $w$, we obtain

$$\max_{1 \leq i \leq n} \|\boldsymbol{a}_i\|_{2k}^{2k} \leq \sum_{j=1}^w \max_{1 \leq i \leq n} \boldsymbol{a}_{ij}^{2k} \leq \frac{2^k B^{k-1} nw}{\eta^k}.$$

Hence, for any $i = 1, 2, ..., n$, $\|\nabla_W f(\mathbf{x}_i, W^*)\|_{2k} \leq \left( \frac{2nw}{B} \right)^{\frac{1}{2k}} \sqrt{\frac{2B}{\eta}}$, which together with Proposition 3 finishes the proof.

### C.2   Proof of Theorem 6

Let $\mathcal{X}_\delta = \bigcup_{i=1}^n \{ \mathbf{x} : \|\mathbf{x} - \mathbf{x}_i\|_2 \leq \delta \}$. Then, we have

$$\mathbb{E}\|f(\mathbf{x}, W^*) - f^*(\mathbf{x})\|_2^2 = \mathbb{E}\left[ \|f(\mathbf{x}, W^*) - f^*(\mathbf{x})\|_2^2 | \mathbf{x} \in \mathcal{X}_\delta \right] + \mathbb{E}\left[ \|f(\mathbf{x}, W^*) - f^*(\mathbf{x})\|_2^2 | \mathbf{x} \notin \mathcal{X}_\delta \right]$$

$$\leq \mathbb{E}\left[ \|f(\mathbf{x}, W^*) - f^*(\mathbf{x})\|_2^2 | \mathbf{x} \in \mathcal{X}_\delta \right] + \varepsilon_2(2M_1^2), \tag{41}$$

where to be short we ignored the subscript $\mathbf{x} \sim \mu$ for the expectations. For any $\mathbf{x} \in \mathcal{X}_\delta$, let $\mathbf{x}_*$ be a training data that satisfies $\|\mathbf{x} - \mathbf{x}_*\| \leq \delta$. By Theorem 5, for any $\mathbf{x} \in \mathcal{X}_\delta$ we have

$$\|\nabla_{\mathbf{x}} f(\mathbf{x}, W^*)\|_{2k} \leq \frac{C \|(W_1^*)^T\|_{2k}}{\min_i \|\mathbf{x}_i\|_{2k}} \left( \left( \frac{2nw}{B} \right)^{\frac{1}{2k}} \sqrt{\frac{2B}{\eta}} + 1 \right). \tag{42}$$

Therefore, by Hölder inequality,

$$|f(\mathbf{x}, W^*) - f^*(\mathbf{x})| \leq |f(\mathbf{x}_*, W^*) - f^*(\mathbf{x}_*)| + \max_{\mathbf{x}' \in \mathcal{X}_\delta} \|\nabla_{\mathbf{x}} f(\mathbf{x}', W^*)\|_{2k} \|\mathbf{x} - \mathbf{x}_*\|_{\frac{2k}{2k-1}}$$

$$+ \max_{\mathbf{x}' \in \mathcal{X}_\delta} \|\nabla_{\mathbf{x}} f^*(\mathbf{x}')\|_2 \|\mathbf{x} - \mathbf{x}_*\|_2$$

$$\leq \frac{C \|(W_1^*)^T\|_{2k}}{\min_i \|\mathbf{x}_i\|_{2k}} \left( \left( \frac{2nw}{B} \right)^{\frac{1}{2k}} \sqrt{\frac{2B}{\eta}} + 1 \right) \sqrt{d} \delta + M_2 \delta. \tag{43}$$

In the last line, the $\sqrt{d}\delta$ term comes from

$$\|\mathbf{x} - \mathbf{x}_*\|_{\frac{2k}{2k-1}} \le \|\mathbf{x} - \mathbf{x}_*\|_2 d^{\frac{k-1}{2k-1}} \le \sqrt{d}\delta.$$

Hence, we have

$$
\begin{aligned}
\mathbb{E}\left[\|f(\mathbf{x}, W^*) - f^*(\mathbf{x})\|_2^2 | \mathbf{x} \in \mathcal{X}_\delta\right] &\le \left[\frac{C\|(W_1^*)^T\|_{2k}}{\min_i \|\mathbf{x}_i\|_{2k}} \left(\left(\frac{2nw}{B}\right)^{\frac{1}{2k}} \sqrt{\frac{2B}{\eta}} + 1\right)\delta + M_2\delta\right]^2 \\
&\le \frac{2dC^2\|(W_1^*)^T\|_{2k}^2}{\min_i \|\mathbf{x}_i\|_{2k}^2} \left(\left(\frac{2nw}{B}\right)^{\frac{1}{2k}} \sqrt{\frac{2B}{\eta}} + 1\right)^2 \delta^2 + 2M_2^2\delta^2.
\end{aligned}
$$

(44)

Inserting (44) to (41) yields the result.

# D Additional proofs

## D.1 Proof of Lemma 1

We show the following more general result.

**Lemma 3.** *Let $A \in \mathbb{R}^{n \times n}$ be a symmetric matrix, and $\mathbf{x} \in \mathbb{R}^n$ be a vector. Then, if $\|A\mathbf{x}\|_2 > \|\mathbf{x}\|_2$, we have*

$$\lim_{m \to \infty} \|A^m \mathbf{x}\|_2 = \infty.$$

The proof is a simple practice for linear algebra. Let $A = Q\Sigma Q^T$ be the eigenvalue decomposition of $A$, and let $\boldsymbol{y} = Q^T \mathbf{x}$. Then, for any $m \in \mathbb{N}^*$ we have

$$\|A^m \mathbf{x}\|_2^2 = \|\Sigma^m \boldsymbol{y}\|_2^2 = \sum_{i=1}^n \sigma_i^{2m} y_i^2,$$

where $\sigma_i$ are eigenvalues of $A$. Hence, $\|A\mathbf{x}\|_2 > \|\mathbf{x}\|_2$ means

$$\sum_{i=1}^n \sigma_i^2 y_i^2 > \sum_{i=1}^n y_i^2,$$

which means there exists $j \in \{1, 2, ..., n\}$ such that $y_j \ne 0$ and $\sigma_i^2 > 1$. Then,

$$\lim_{m \to \infty} \|A^m \mathbf{x}\|_2^2 = \lim_{m \to \infty} \sum_{i=1}^n \sigma_i^{2m} y_i^2 \ge \lim_{m \to \infty} \sigma_j^{2m} y_j^2 = \infty.$$

## D.2 Proof of Lemma 2

When $t \in [0, 1)$, we have $t^k + (1-t)^k \le (t + 1 - t)^k = 1$. Hence,

$$t^k \le 1 - (1-t)^k \le 1 + (t-1)^k \le 2^{k-1}((t-1)^k + 1).$$

When $t \ge 1$, by the Hölder inequality,

$$t = (t-1) + 1 \le \left((t-1)^k + 1\right)^{\frac{1}{k}} (1+1)^{1-\frac{1}{k}}.$$

Taking $k$-th order on both sides, we have

$$t^k \le 2^{k-1}((t-1)^k + 1).$$

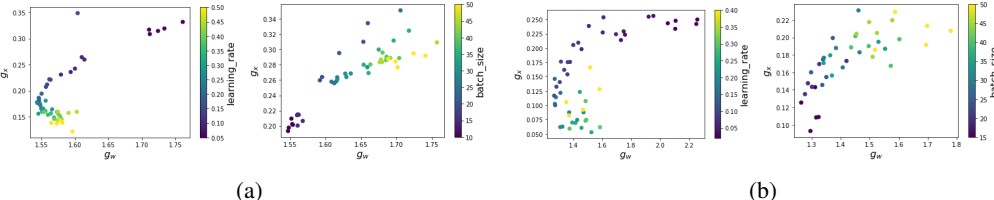

(a)                     (b)

Figure 2: Results for $g_W^k$ and $g_{\mathbf{x}}^k$ with $k = 2$, on a fully-connected neural network trained on FashionMNIST (shown in (a)) and a VGG-11 network trained on CIFAR10 (shown in (b)). For both (a) and (b), the left panel shows $g_W^k$ and $g_{\mathbf{x}}^k$ of the solutions found by SGD with different learning rate, while batch size fixed at 20. The right panel shows solutions found by SGD with different batch size, with learning rate fixed at 0.1.

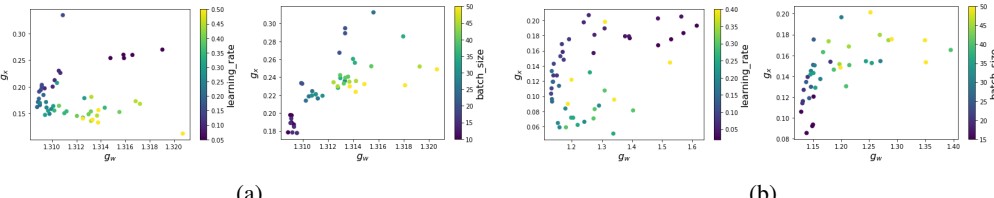

(a)                     (b)

Figure 3: Results for $g_W^k$ and $g_{\mathbf{x}}^k$ with $k = 3$, on a fully-connected neural network trained on FashionMNIST (shown in (a)) and a VGG-11 network trained on CIFAR10 (shown in (b)). For both (a) and (b), the left panel shows $g_W^k$ and $g_{\mathbf{x}}^k$ of the solutions found by SGD with different learning rate, while batch size fixed at 20. The right panel shows solutions found by SGD with different batch size, with learning rate fixed at 0.1.

## E Additional Experiments

Except the $g_W$ and $g_{\mathbf{x}}$ in (5), we also checked the gradient norms with higher $k$ in the same experiment settings. We consider

$$g_W^k := \left( \frac{1}{n} \sum_{i=1}^{n} \|\nabla_W f(\mathbf{x}_i, W)\|_{2k}^{2k} \right)^{\frac{1}{2k}} , \quad g_{\mathbf{x}}^k := \left( \frac{1}{n} \sum_{i=1}^{n} \|\nabla_{\mathbf{x}} f(\mathbf{x}_i, W)\|_{2k}^{2k} \right)^{\frac{1}{2k}} . \quad (45)$$

Figures 2 and 3 show the scatter plot for $k = 2$ and $k = 3$, respectively. The figures shows that in most cases there is still a strong correlation between the gradient norm with respect to $W$ and the gradient norm with respect to $\mathbf{x}$.

The next figure (Figure 4) shows experiment results on more complicated dataset and network architectures, where we trained a 14-layer Resnet on a subset of the CIFAR100 dataset. The scattered plots show similar results as in Figrue 1, which further justifies our theoretical predictions.

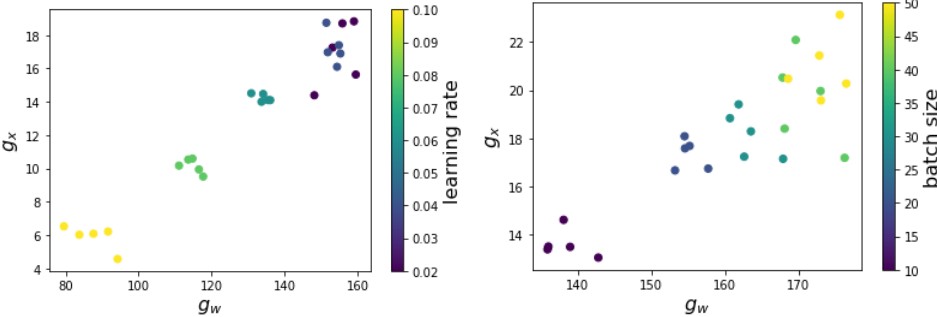

Figure 4: Results for a Resnet trained on CIFAR100 dataset. (Left) $g_W$ and $g_{\mathbf{x}}$ of the solutions found by SGD with different learning rate, while batch size fixed at 20. (Right) $g_W$ and $g_{\mathbf{x}}$ of the solutions found by SGD with different batch size, with learning rate fixed at 0.05.

# F  Experiment details

**General settings**   In the numerical experiments shown by Figure 1, 2 and 3, we train fully-connected deep neural networks and VGG-like networks on FashionMNIST and CIFAR10, respectively. As shown in the figures, for each network, different learning rates and bacth sizes are chosen. 5 repetitions are conducted for each learning rate and batch size. In each experiment, SGD is used to optimize the network from a random initialization. The SGD is run for 100000 iterations to make sure finally the iterator is close to a global minimum. then, $g_{\mathbf{x}}$ and $g_W$ in (5) are evaluated at the parameters given by the last iteration. In the experiments shown in Figure 4, we train a residual network on CIFAR100. Experiments are still repeated by 5 times in each combination of learning rate and batch size. In each experiment, SGD is run for 50000 iterations. All experiments are conducted on a MacBook pro 13" only using CPU. See the code at `https://github.com/ChaoMa93/Sobolev-Reg-of-SGD`.

**Dataset**   For the FashionMNIST dataset, 5 out of the 10 classes are picked, and 1000 images are taken for each class. For the CIFAR10 dataset, the first 2 classes are picked with 1000 images per class. For the CIFAR100 dataset, the first 10 classes for picked with 500 images in each class.

**Network structures**   The fully-connected network has 3 hidden layers, with 500 hidden neurons in each layer. The ReLU activation function is used. The VGG-like network consists of a series of convolution layers and max pooling layers. Each convolution layer has kernel size $3 \times 3$, and is followed by a ReLU activation function. The max poolings have stride 2. The order of the layers are

$$16-> M -> 16 -> M -> 32 -> M -> 64 -> M -> 64 -> M,$$

where each number means a convolutional layer with the number being the number of channels, and "M" means a max pooling layer. A fully-connected layer with width 128 follows the last max pooling.

The residual network takes conventional architecture of Resnet. It consists of 6 residual blocks. The number of channels in the blocks are $32, 32, 64, 64, 128, 128$, from the input block to the output block.