# OpenReview forum: "On Linear Stability of SGD and Input-Smoothness of Neural Networks"
_NeurIPS.cc/2021/Conference — NeurIPS 2021 Spotlight_

### Official Review · Reviewer_gdzy · 2021-07-11

**Rating:** 6
**Confidence:** 4

**Summary:**

Prior work on minima stability for SGD considered linear stability in expectation using second order moment. This paper considers linear stability in higher moments, while providing necessary and sufficient conditions for it. Using these conditions and the multiplicative structure of neural networks, the authors show that the Sobolev semi-norm of SGD solutions (w.r.t. the input) is bounded from above. Then, under certain smoothness assumptions on the learned function and the target function, the authors prove generalization bounds and adversarial robustness.

**Limitations And Societal Impact:**

See above

**Main Review:**

The article is well written, and the proofs are rigorous. I carefully checked the proofs of Theorems 1-3, and they look fine. On the other proofs I went over more generally, and they also seem good.

Here are my comments:
1.	Why should we consider higher moments for linear stability? E.g., why second order is not sufficient to say that a minimum is stable? If a minimum is not stable for some order but is stable in a lower order, what this tells us? Can a minimum be stable for any order? if so, does it have special properties?
2.	The bounds proved by the multiplicative property, namely Theorem 4-7, can be loose. For example, consider the simple case of a two-layer univariate network with ReLU activation (i.e. scalar input/output with a wide hidden layer). Since ReLU is positive homogenous we can arbitrarily increase the $ \ell_p $-norm of the first layer while preserving the function implemented by the network, and thus making the bounds trivial. This is true for any network with ReLU activation (or its variants).
3.	From 2., it is not true to say that SGD implicitly regulate the Sobolev semi-norm of the function implemented by the network, since there is no (proven) mechanism in SGD that keeps the the norm of the first layer small.
4.	The authors claim that Corollary 3 is more accurate than previous known result, which is written in Proposition 1 (see row 235), however I did not see any explanation nor proof for this claim (maybe I have missed it...).
5.	Most of the results hold only for the quadratic loss.
6.	I think that Lemma 1 is used to prove necessity in the proof of Theorem 1, and not sufficiency.


**Time Spent Reviewing:**

10 Hours

---

> ### Author Response · Authors · 2021-08-10
> **Response to Reviewer gdzy**
>
> We thank the reviewer for the comments, and for carefully checking the proof. We answer the reviewer's questions in the following.
>
> **1. Why should we consider higher moments for linear stability? E.g., why second order is not sufficient to say that a minimum is stable? If a minimum is not stable for some order but is stable in a lower order, what this tells us? Can a minimum be stable for any order? if so, does it have special properties?**
>
> **response**: There are at least two benefits to consider the stability of higher moments. First, the stability of higher moments is stronger than the stability of lower moments, instead of being equivalent. For a minimum that is stable for higher moments, the distribution of the iterators is more concentrated around the global minimum. For example, if the minimum is second order stable, the second moment of the iterator's distribution is bounded, but its higher moments can go to infinity, and the trajectory of iterators may have an unbounded subsequence. If a minimum is stable for any order, then the trajectory lies in a bounded set. Second, by Theorem 3, 4, and 6, higher order stability leads to better bounds on Sobolev seminorm and generalization error. Especially, the order of stability influences the order of $n$ in the generalization error, which is a crucial term in the bounds. With a fixed data distribution, the generalization error decreases faster when the minimum is stable with higher order moments.
>
> **2. The bounds proved by the multiplicative property, namely Theorem 4-7, can be loose. For example, consider the simple case of a two-layer univariate network with ReLU activation (i.e. scalar input/output with a wide hidden layer). Since ReLU is positive homogenous we can arbitrarily increase the $l_p$-norm of the first layer while preserving the function implemented by the network, and thus making the bounds trivial. This is true for any network with ReLU activation (or its variants).**
>
> **response**: By the homogeneity of ReLU activation, one can indeed make the input layer parameters arbitrarily large while not changing the function implemented by the model. However, this is unlikely to happen in practice, especially for the trajectories produced by SGD. For ReLU network trained with SGD, there is an invariant involving weights from neighboring layers [1]. This guarantees that the weights from all layers get big or small at the same time. As long as the initialization is good, it is unlikely that one layer becomes very big while another becomes very small. Please also see our response to Reviewer 3 for more discussion on this issue, as well as some numerical results.
>
> **3. From 2, it is not true to say that SGD implicitly regulate the Sobolev semi-norm of the function implemented by the network, since there is no (proven) mechanism in SGD that keeps the the norm of the first layer small.**
>
> **response**: By the discussion for point (2), at least for ReLU network, SGD can indeed keep the input layer parameters from getting excessively large. Yet, we agree that before obtaining a rigorous proof of SGD's control on first-layer parameters, it is better to say SGD stability and small first layer parameters regulate the Sobolev semi-norm together. Within the two factors, we want to emphasize the significant role played by SGD stability. To be precise, we will change the paper's title to "The Connection Between Linear Stability of SGD and the Smoothness of Neural Networks as a Function of Input Data".
>
> **4. The authors claim that Corollary 3 is more accurate than previous known result, which is written in Proposition 1 (see row 235), however I did not see any explanation nor proof for this claim (maybe I have missed it...).**
>
> **response**: To avoid overclaiming, we will change the sentence "we can write down a more accurate sufficient condition for linear stability than Proposition 1" to "we can write down another sufficient condition for linear stability". Though, we believe the condition in Corollary 3 is more accurate than that in Proposition 1, because the former one is derived directly from a closed dynamics of $\mathbb{E} \tilde{W}_t^{\otimes 2}$. We will leave the study of the two sufficient conditions' relation to future work.
>
> **5. Most of the results hold only for the quadratic loss.**
>
> **response**: We agree that most of our analysis are conducted on square loss. We believe that similar results will hold on other strongly convex losses, which have no essential difference from the square loss. Though, our analysis may not directly apply on the cross entropy loss. The biggest difference for the cross entropy loss is that the global minima of the cross entropy loss are usually infinitely big. Hence, one have to consider directions as minimum instead of points. We leave detailed analysis as future work.
>
> **6. I think that Lemma 1 is used to prove necessity in the proof of Theorem 1, and not sufficiency.**
>
> **response**: We thank the reviewer for pointing this out. Indeed, Lemma 1 is only used in the proof  ofnecessity. In the revised version of the paper, we will move this Lemma after the proof of sufficiency.
>
> [1] Arora, Sanjeev, Nadav Cohen, and Elad Hazan. "On the optimization of deep networks: Implicit acceleration by overparameterization." International Conference on Machine Learning. PMLR, 2018.

---

> > ### Comment · Reviewer_gdzy · 2021-08-29
> > **Response to Authors**
> >
> > Thanks for the detailed rebuttal.

---

### Official Review · Reviewer_PF9o · 2021-07-14

**Rating:** 8
**Confidence:** 4

**Summary:**

The paper extends previous stability analysis ([30]) to high-order moments of the iterator. They characterize the implicit regularization of SGD by showing that a stable global minimum must satisfy conditions regarding its flatness and non-uniformity. In addition, the authors build relations between the model function’s derivative w.r.t. the parameters and w.r.t. to the inputs. Using these relations, they provide upper bounds for the Sobolev seminorm of the model function at stable SGD interpolating solutions. In addition, under additional smoothness conditions, the authors were able to extend that last result to a neighborhood of the training data. Finally, the authors built upon the Sobolev regularization and use it to provide bounds on the generalization performance and adversarial robustness.

**Ethical Concerns:**

None.

**Limitations And Societal Impact:**

The authors thoroughly discussed the limitations of their results.

**Main Review:**

This paper examines theoretically how does SGD selects the global minima it converges to. Specifically, the authors analyze the implicit regularization effect and generalization performance of the solution found by SGD using a linear stability analysis.

The paper main contributions are:
-	The authors prove upper bounds on the gradients’ moment at the solution found by SGD, assuming this solution is stable. These conditions also include control on the flatness and non-uniformity. This result provides a better characterization of the implicit bias of SGD.
-	The authors provide upper bounds for the Sobolev seminorm of the model function at stable SGD interpolating solutions. This implies that the Sobolev seminorms of the model on the training data are regularized by the linear stability. In addition, under some smoothness assumption, this result is extended to a neighborhood of the training data. These results suggest a possible explanation for the good generalization of flat minima.
-	Under additional assumptions on the data distribution, the authors provide bounds on the generalization error and adversarial robustness.

Clarity/writing:

I enjoyed reading the paper. The paper is well written, the assumptions and results are clearly stated, and many explanations were provided throughout the paper.

My main question:
-	Many of the upper bounds suggested in the paper include the norm of the first layer weights. Can the author extend on the assumption that the input layer weights are not very large?

Minor comments:
-	There is a typo in the title (Stocahstic --> Stochastic)
-	I think that adding “sharpness” and “non-uniformity” as formal definitions will make the paper a bit clearer (these terms were explained verbally but it is easy to miss this)

---

Update: Thank you for the detailed rebuttal and for elaborating on the upper limit with experimental results. This helped to elevate my concern.


**Time Spent Reviewing:**

11

---

> ### Author Response · Authors · 2021-08-10
> **Response to Reviewer PF9o**
>
> We thank the reviewer for appreciating our work, as well as the valuable suggestions and comments. We will correct the typo on the title, and add the definition of sharpness and non-uniformity to make the presentation clearer, as suggested by the reviewer.
>
> The main question: Many of the upper bounds suggested in the paper include the norm of the first layer weights. Can the author extend on the assumption that the input layer weights are not very large?  **Answer:** We conducted preliminary experiments concerning the first layer weights. Results show that usually the first layer weights do not get very big. It increases at the early phase of the training, but gets stable afterwards (e.g. after $20000$ iterations). Please see the following table for some results.
>
> |n_iters | 0 | 100 | 1000 | 10000 | 20000 | 50000 |
> |----------|---|----------|--------|----------|----------|----------|
> |$\|\|W_1\|\|_2$ | 1.032 | 1.034 | 1.218 | 2.011 | 2.257 | 2.270 |
>
> The table shows the norm of first layer parameter matrix at different iterations. We train a fully-connected neural network on FashionMNIST. The experiment settings are the same as the experiments in the paper. The results are averages from 5 runs.
> We will do more experiments and add some discussion on this issue in the revised version of the paper. Theoretical insights also exist. For example, for fully-connected ReLU networks trained with SGD there is an invariant between parameters from neighboring layers (Similar to that in [1]). This imposes a balance between all the layers and prevents parameters of a single layer from getting too big.
>
> [1] Arora, Sanjeev, Nadav Cohen, and Elad Hazan. "On the optimization of deep networks: Implicit acceleration by overparameterization." International Conference on Machine Learning. PMLR, 2018.

---

### Official Review · Reviewer_Ggg9 · 2021-07-20

**Rating:** 7
**Confidence:** 3

**Summary:**

This paper shows that due to the multiplicative structure of input data and first layer parameters in a MLP neural network the flat minima regularizes the norm of the gradient of the model and then explains its generalizability. It also shows that SGD actually regularizes higher-order moments of the gradient of the noise when it is close to an interpolating minima. Utilizing this property of SGD and multiplicative structure shows that SGD minimizes the Sobolev seminorms of the model. The presented analysis is based on quadratic estimation of the objective around the minima and uses linear stability analysis.

**Limitations And Societal Impact:**

Yes

**Main Review:**

Understanding the implicit regularization imposed by SGD has great importance in understanding the generalization performance in deep neural networks. This paper utilizes the product structure between the first layer parameter and the input data and discovers the relation between flat minima and the gradient norm or the Lipschitz constant of the model that improves the stability of the model.

However, the major theoretical contribution of the paper is local and based on a quadratic approximation of the model around an interpolating local minima. The major part of the analysis is based on the existing results in the literature for quadratic models.
Besides, it makes a bunch of assumptions and it does not explain which of them are true in a multi-layer perceptron model.

In terms of the write-up, there are a bunch of typos here and there that need to be fixed such as stocahstic instead of stochastic in the title.

My question


1- It is not clear how restrictive the assumptions the paper makes. It would be clarifying to explain if these assumptions are true for deep neural networks.

2- The product structure has been shown that holds for MLP. Does it hold for other architectures such as CNN or transformers?

3- In practice we usually normalize the data, therefore some x_i could be very close to zero. Based on this, how tight do you think the upper bound in 4 could be?

4- You assume that SGD with fixed step size would converge to an interpolating solution. It would be helpful to explain under which condition this can happen and why you think MLP satisfies these requirements.


**Time Spent Reviewing:**

5

---

> ### Author Response · Authors · 2021-08-10
> **Response to Reviewer Ggg9**
>
> We thank the reviewer for the valuable questions and comments. We apologize for the confusion caused by the typos, especially the one on the title. We will carefully read through the whole paper and correct all typos.
>
> For the questions raised by the reviewer:
>
> **1. It is not clear how restrictive the assumptions the paper makes. It would be clarifying to explain if these assumptions are true for deep neural networks.**
>
> **response**: Our assumptions lie in two categories: assumptions on data and assumptions on networks. We have one assumption on neural networks: the local smoothness condition (definition 1). This condition is actually weaker than second-order smoothness, since we only require the gradient (with respect to $W$) be bounded around the global minimum. Therefore, this assumption holds locally in a neighborhood around the global minimum as long as the network function is second-order differentiable. We do not provide the size of the neighborhood, since there is no good tool to derive non-vacuous bounds for neural networks' derivatives.
>
> The other assumptions like those in Definition 2, 3, and 4 are for the distribution of input data. These assumptions usually hold for distribution supported on low-dimensional surfaces. Practical datasets for neural networks are widely believed to lie on low dimensional sets. Works have been done to explore the intrinsic dimension of image datasets [1]. However, since we do not know the exact distribution of real data, it is not easy to precisely verify the assumptions made in our paper on real data.
>
> **2.The product structure has been shown that holds for MLP. Does it hold for other architectures such as CNN or transformers?**
>
> **response**: The product structure holds as long as the input vector is first multiplied with a parameter matrix before any other nonlinear operation. It holds for nearly all popular network structures. A convolution is a linear transformation for tensors. If we vectorize the input images, then the convolution becomes a product of the vectorized input with a parameter matrix. The parameter matrix has some special structures, such as parameter sharing. But the product structure between the input data and the parameter matrix still holds. This is also true for ResNets, DenseNets, RNNs (LSTMs), and even Transformers, which all involves a direct linear transformation on the input data (in the forms of simple linear transformations, convolutions or attentions). Besides, experimental results in the paper (on CIFAR10 and CIFAR100) also show that our results hold numerically for CNN.
>
>
> **3.In practice we usually normalize the data, therefore some $x_i$ could be very close to zero. Based on this, how tight do you think the upper bound in 4 could be?**
>
> **response**: If the normalization is done element-wise, then it is possible that some data will become very small after normalization. These data have close-to-average values on every entry. These data can make some of our bounds loose, such as the bounds in Theorem 3, 4, and 6. However, such small data is only a very small portion of all data, and the portion gets exponentially smaller as the dimension increases. Therefore, when the input dimension is big, we can just ignore these data in our bounds, and only consider data whose norms are big enough. For example, in Theorem 6 we can put the small data (and their neighborhood) into the last term in the bound, which characterizes a small portion of space that is not covered by the estimate of Sobolev seminorm.
>
> If the normalization is done point-wise, i.e. normalize each data point to have norm 1 (this is assumed in many theoretical work), then the normalization will not have negative impact to our bounds. On the contrary, it prevents the data from being too small.
>
>
> **4.You assume that SGD with fixed step size would converge to an interpolating solution. It would be helpful to explain under which condition this can happen and why you think MLP satisfies these requirements.**
>
> **response**: There are both numerical and theoretical evidence that over-parameterized neural networks can fit all the training data and find interpolating solutions during training. Numerical observations that neural network can reduce training error to zero are provided in many works, such as [2]. Theoretical analysis are provided in the highly over-parameterized cases, such as the Neural Tangent Kernel (NTK) regime [3]. The puzzle is that neural networks can still generalize well when they are over-parameterized and can easily fit all the training data. Our work provide an explanation to this puzzle by identifying an implicit regularization effect of SGD.
>
>
> [1] Pope, Phillip, et al. "The Intrinsic Dimension of Images and Its Impact on Learning." arXiv preprint arXiv:2104.08894 (2021).
>
> [2] Zhang, Chiyuan, et al. "Understanding deep learning (still) requires rethinking generalization." Communications of the ACM 64.3 (2021): 107-115.
>
> [3] Jacot, Arthur, Franck Gabriel, and Clément Hongler. "Neural tangent kernel: Convergence and generalization in neural networks." arXiv preprint arXiv:1806.07572 (2018).

---

### Official Review · Reviewer_iuXC · 2021-07-20

**Rating:** 6
**Confidence:** 4

**Summary:**

This work presents a novel framework of "Sobolev regularization" effect of stochastic gradient descents for quantifying and establishing wideness of a global interpolating optimum in SGD. The key idea is that because the neural network input $x$ is multiplids to weights $W_1$ and then provided into the neural network, regularity properties of the weights near the interpolating global optimum imply regularity properties (wideness) of the minimum.

**Main Review:**

Overall, I feel that the viewpoint presented in this work is novel and worthy of publication. The key insight is that since the input $x$ and the neural network weights are first multiplied and then entered into the network, so regularity (stability) properties on $W$ can be transferred onto regularity (wideness/small Sobolev norm) properties with respect to $x$ is a very nice one.

I have one not-so-minor comment. Definition 5 seems to be incomplete; the relation between $W^*$ and $\tilde{W}^{\otimes k}_t$ is not completely specified. (All that is said is that $\{W_t\}$ is close to $W^*$.) I think the issue is that the authors have the mental model of an interpolating global minimum as an isolated minimum. However, there is some recent evidence that a connected set of global minima is a more accurate model.
https://proceedings.neurips.cc/paper/2018/hash/be3087e74e9100d4bc4c6268cdbe8456-Abstract.html
http://proceedings.mlr.press/v119/shevchenko20a.html
Can the authors comment on how having a connected set of interpolating global minima may or may not affect the analysis?

**Time Spent Reviewing:**

4

---

> ### Author Response · Authors · 2021-08-10
> **Response to Reviewer iuXC**
>
> We thank the reviewer for the appreciation of the insight provided in our paper. For the reviewer's question on Definition 5, we would like to point out that this definition only involves the linearized dynamics of $\tilde{W}_t$, and therefore we do not need to assume $\tilde{W}_0$ is close to $W^*$. An issue is how well is the quadratic approximation to the actual landscape, and equivalently how well does the linearized dynamics approximate the real dynamics. We believe this is also the reviewer's main problem. We did not provide an estimate of the size of the neighborhood in which this approximation works well. Though, by the Taylor's theorem, quadratic approximation always holds locally as long as the loss function is twice differentiable, no matter it is locally convex or not. Therefore, even for the case where interpolating global minima are connected and hence local landscape is nonconvex, around a sufficiently small neighborhood of (any) global minima the quadratic approximation of the loss function still holds. As a consequence, any optimization dynamics that converges to a global minima $W^*$ has to satisfy the linear stability condition given in Definition 5 (with respect to this specific $W^*$).

---

### Decision · Program_Chairs · 2021-09-27

**Decision:**

Accept (Spotlight)

**Comment:**

All reviewers have praised the clarity of the paper as well as the soundness and novelty of the results. Some important questions regarding completeness of some definitions and applicability of the current assumptions have been raised in the reviews, which the authors have clarified with a strong rebuttal. I encourage the authors to incorporate these remarks in the camera ready version.